# Unique trackway on Permian Karoo shoreline provides evidence of temnospondyl locomotory behaviour

**David P. Groenewald**[1¤]*, **Ashley Krüger**[2], **Michael O. Day**[1,3], **Cameron R. Penn-Clarke**[1], **P. John Hancox**[1], **Bruce S. Rubidge**[1]

**1** Evolutionary Studies Institute, University of the Witwatersrand, Johannesburg, South Africa, **2** Department of Palaeobiology, Swedish Museum of Natural History, Stockholm, Sweden, **3** Fossil Reptiles, Amphibians and Birds Section, Natural History Museum, London, United Kingdom

¤ Current address: Institut Català de Paleontologia Miquel Crusafont, Cerdanyola del Vallès, Spain
* david.groenewald@wits.ac.za

**Data Availability Statement:** All relevant data are within the paper and its Supporting information files. Three-dimensional models have been uploaded to the MorphoSource data archive:

## Abstract

Large-bodied temnospondyl amphibians were the dominant predators in non-marine aquatic ecosystems from the Carboniferous to the Middle Triassic. In the Permian-aged lower Beaufort Group of the main Karoo Basin, South Africa, temnospondyls are represented exclusively by the family Rhinesuchidae and are well represented by body fossils, whereas trace fossils are scarce. Accordingly, most interpretations of the behaviour of this family are based on skeletal morphology and histological data. Here we document the sedimentology and palaeontology of a late Permian palaeosurface situated immediately below the palaeoshoreline of the Ecca Sea (transition from the Ecca Group to the Beaufort Group) near the town of Estcourt in KwaZulu-Natal Province. The surface preserves numerous ichnofossils, including tetrapod footprints and fish swim-trails, but most striking are seven body impressions and associated swim trails that we attribute to a medium-sized (~1.9 m long) rhinesuchid temnospondyl. These provide valuable insight into the behaviour of these animals. The sinuous shape of some of the traces suggest that the tracemaker swam with continuous sub-undulatory propulsion of the tail.

## Introduction

The main Karoo Basin is renowned for its rich vertebrate body fossil record, as well as its extensive ichnological record comprising terrestrial and aquatic traces from the Upper Pennsylvanian to the Lower Jurassic. A diverse array of trace fossils has been recognised with attributed trace- and trackmakers including fish [e.g. 1, 2], invertebrates [e.g. 3–10], therapsids [9, 11–16], dinosaurs and other reptiles [16–27], and amphibians [21, 28–30].

A large palaeosurface with several remarkable trace fossils is exposed along the course of a tributary of the Rensburgspruit in the uThukela District, KwaZulu-Natal Province. The most striking ichnofossils are of seven large (>1 m long) impressions recording resting and locomotor behaviours. While the unique morphology of these traces and the importance of the site

https://www.morphosource.org/projects/
000494771?locale=en.

**Funding:** Financial support for this project was
provided by the National Research Foundation
(NRF) and its African Origins Platform, GENUS (the
DSI-NRF Centre of Excellence in Palaeosciences),
and the Palaeontological Scientific Trust (PAST).
DPG received funding from the European Union's
Horizon Europe research and innovation
programme under the Marie Skłodowska-Curie
actions (grant agreement: 101060666) when
revising the manuscript. The funders had no role in
study design, data collection and analysis, decision
to publish, or preparation of the manuscript.

**Competing interests:** The authors have declared
that no competing interests exist.

has previously been recognised [e.g. 31], neither the site nor the trackways have been fully
described. This is mainly because the size (> 1 m) and shallow depth (< 5 mm) of the impres-
sions resulted in traditional casting methods being unsuccessful. Other traces preserved on the
palaeosurface include numerous smaller (10–15 cm diameter) subcircular to blob-shaped
depressions and paired and unpaired linear traces.

In this paper, we document the site with its sedimentary structures and unique traces as
well as its stratigraphic context. Using high-resolution three-dimensional (3D) surface scans
and aerial orthophotographs, we provide the first comprehensive description of the large
impressions. A probable tracemaker for the impressions is assigned and locomotory behaviour
of the tracemaker is inferred.

## Geological and palaeontological context

**Geological background and stratigraphy of the palaeosurface.** Named in honour of the
late Mr Dave Green, who discovered this remarkable site and had a passion for palaeontology,
the Dave Green palaeosurface (S28.967122˚, E29.987366˚) is located along a tributary of the
Rensburgspruit on the farm Van der Merwe's Kraal 972, approximately 10 km northeast of the
town Estcourt in KwaZulu-Natal Province, South Africa (Figs 1 and 2). The region spans the
Ecca-Beaufort transition and is primarily underlain by late Permian (Changhsingian) siliciclas-
tic deposits of the Waterford and Balfour formations that are intruded by lower Jurassic doler-
ite dykes and sills [32–37]. The Dave Green palaeosurface covers an area of approximately 600
m$^2$ and is situated in the upper Waterford Formation, immediately below the Ecca-Beaufort
contact, which records the transition from marine/ lacustrine to fluvial environmental condi-
tions [31, 34–37].

In the southern Karoo Basin, the Waterford Formation constitutes the uppermost Ecca
Group [38–43], but has recently been shown to also be present in the north of the basin, where
it gradationally overlies the Volksrust or Tierberg formations [37]. This diagnostic and easily
recognisable formation, which incorporates strata previously assigned to the upper Volksrust
and/or conformably overlying lower Balfour Formation (including the now defunct Norman-
dien and Estcourt formations [36, 37]), is characterized as being generally arenaceous in
nature, generally ripple-marked, as well as containing rhythmically-bedded carbonaceous
shale and ubiquitous soft sediment deformation structures (Fig 3).

Previous studies showed that these deposits accumulated in a regressive delta front and
delta plain [34, 37, 41, 43–46]. Within the study area, the Waterford Formation is at least 140
m thick and is considered to be Wuchiapingian in age [36]. Mudstones of the Waterford For-
mation are commonly dark to medium grey (N3 –N5) and Olive Green (5GY 3/2), whereas
siltstones and sandstones are typically lighter, varying between medium light grey to light grey
(N6 –N8), and occasionally light brown (5YR 6/4). Laminated mudstones typically display
platy/ flaky weathering and may contain fragmentary plant impressions and invertebrate
traces.

The overlying Balfour Formation is, with respect to the Waterford Formation, markedly
more argillaceous, with interspersed sandstone lenses that typically have an erosional base. In
contrast to the dark grey to olive-green mudstones of the underlying Waterford Formation,
mudstones of the Balfour Formation are generally moderate yellow (5Y 7/6) to dark greenish
yellow (10Y 6/6), are massive to finely-laminated, display a blockier weathering pattern, and
may contain calcareous nodules and slickensided surfaces (Fig 3). Plant impressions and tetra-
pod vertebrate remains are more abundant and complete in the Balfour Formation than the
underlying Waterford Formation. The depositional environment for the Balfour Formation in

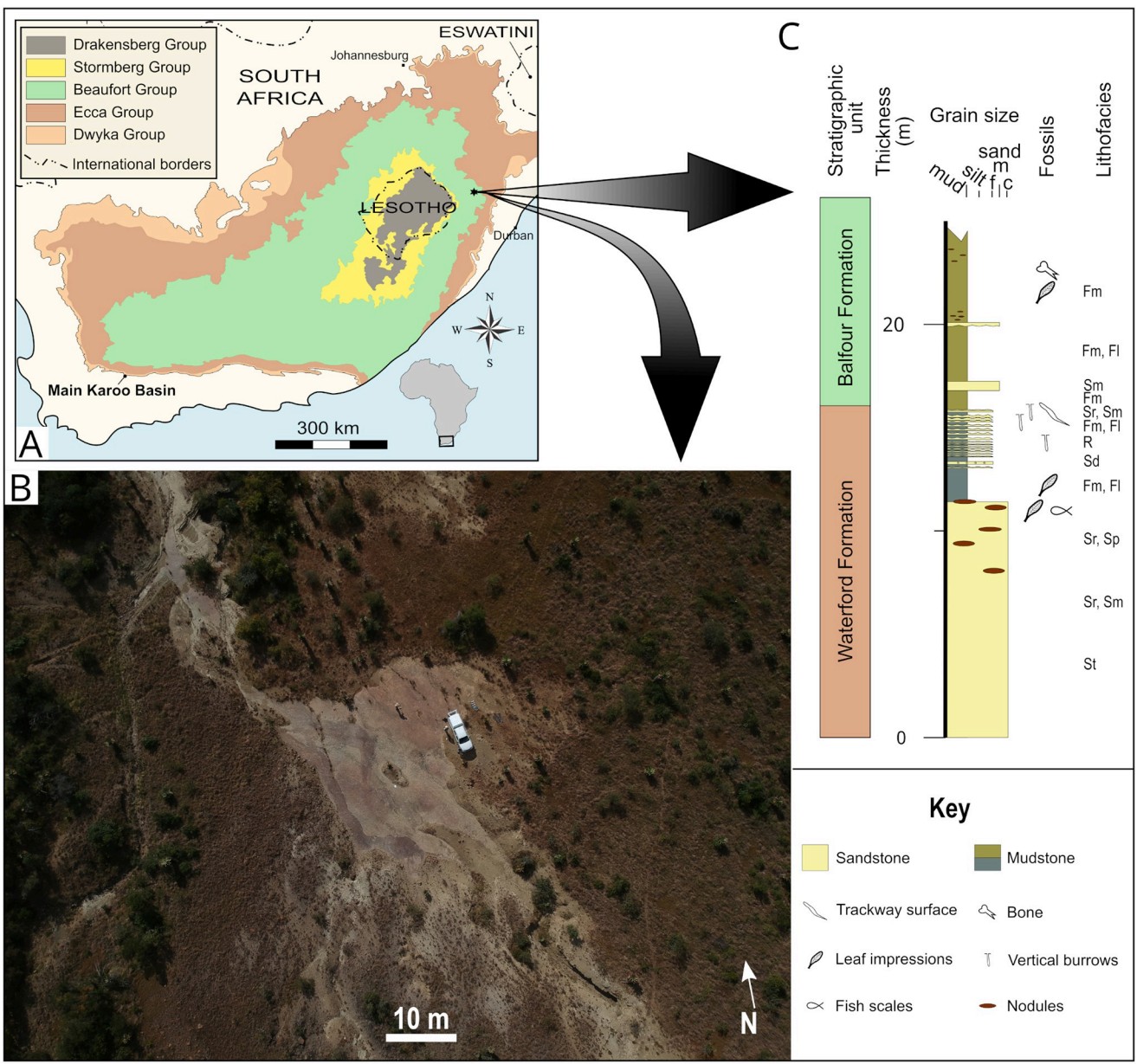

**Fig 1. Geological setting of the Dave Green palaeosurface.** A) Simplified geological map of the main Karoo Basin. Position of the study area is indicated. B) Aerial photo of the palaeosurface (taken by AK). C) Stratigraphic log measured along the Rensburgspruit.

the northeastern main Karoo Basin has been interpreted as having accumulated in high-load meandering river systems [34, 36, 37, 39, 44, 47–51].

## Local palaeontology

Van der Merwe's Kraal 972, as well its immediate surrounds, is palaeontologically rich, comprising a diverse assemblage of plant, insect, and vertebrate body fossils in addition to trace fossils [31, 34, 52–56]. Green [34] undertook detailed sedimentological and palaeontological work on the farm but although she noted and briefly discussed the Dave Green palaeosurface in her "shoreline log", she largely ignored the tetrapod traces. Green (1997) did, however,

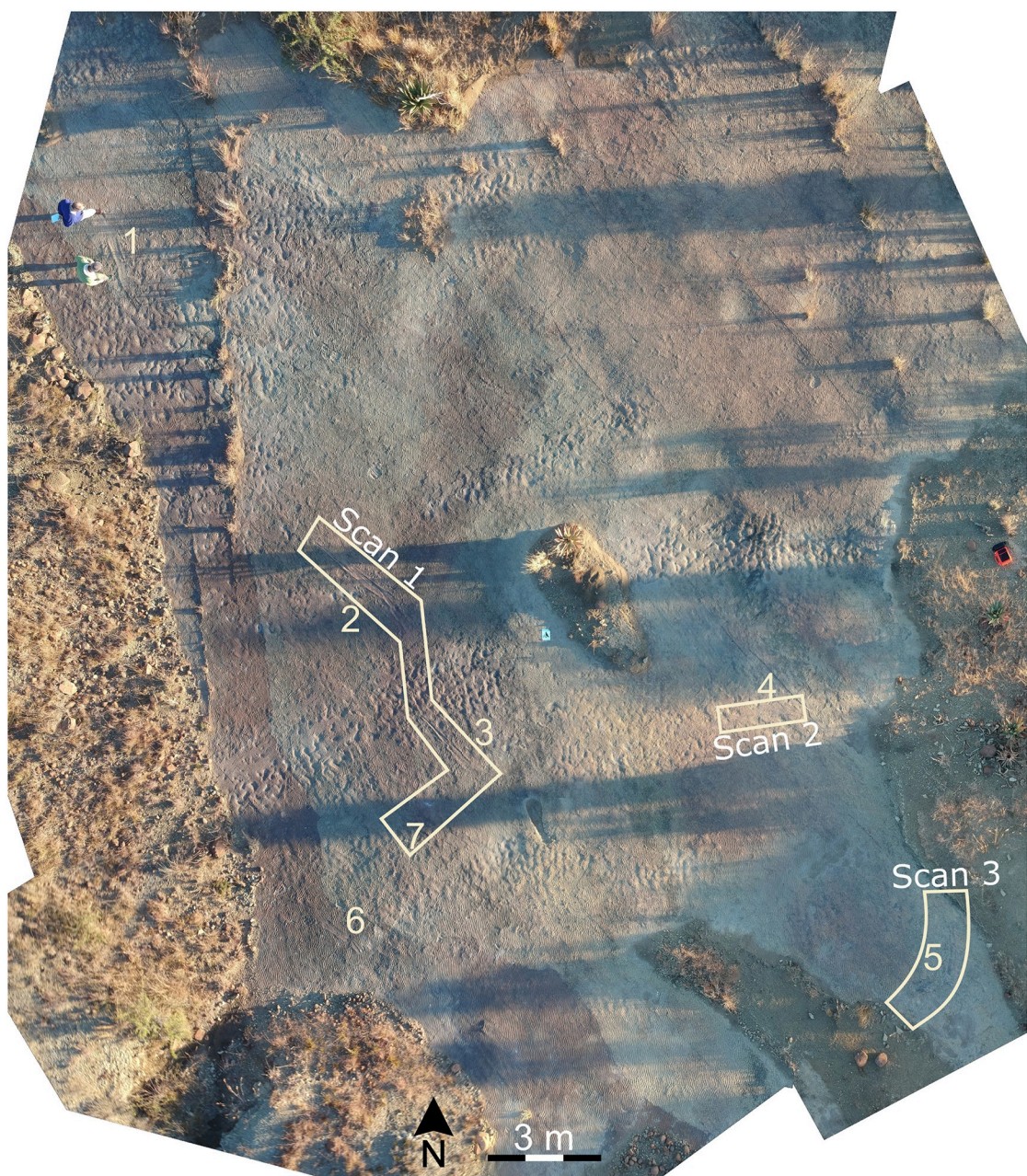

**Fig 2. Orthophoto of the palaeosurface with the positions of the seven large impressions and paths followed for the high-resolution scans indicated.** Numbers used for the impressions are the same throughout the text.

document 15 trackways from two surfaces roughly 1.1 km to the south of the Dave Green palaeosurface. These surfaces, which have since been re-covered by silt and mud, had been exposed during the excavation of a dam and Green [34] considered them to be from a laterally equivalent stratigraphic level as the Dave Green palaeosurface. The 15 trackways were attributed to small-to-medium sized dicynodont trackmakers, based on track morphology and because dicynodonts are the most commonly represented group in the body fossil record, and interpreted as reflecting undertracks and movement of an animal buoyed by water [34]. Plant fossils found near the Dave Green palaeosurface include impressions of *Glossopteris* leaves and

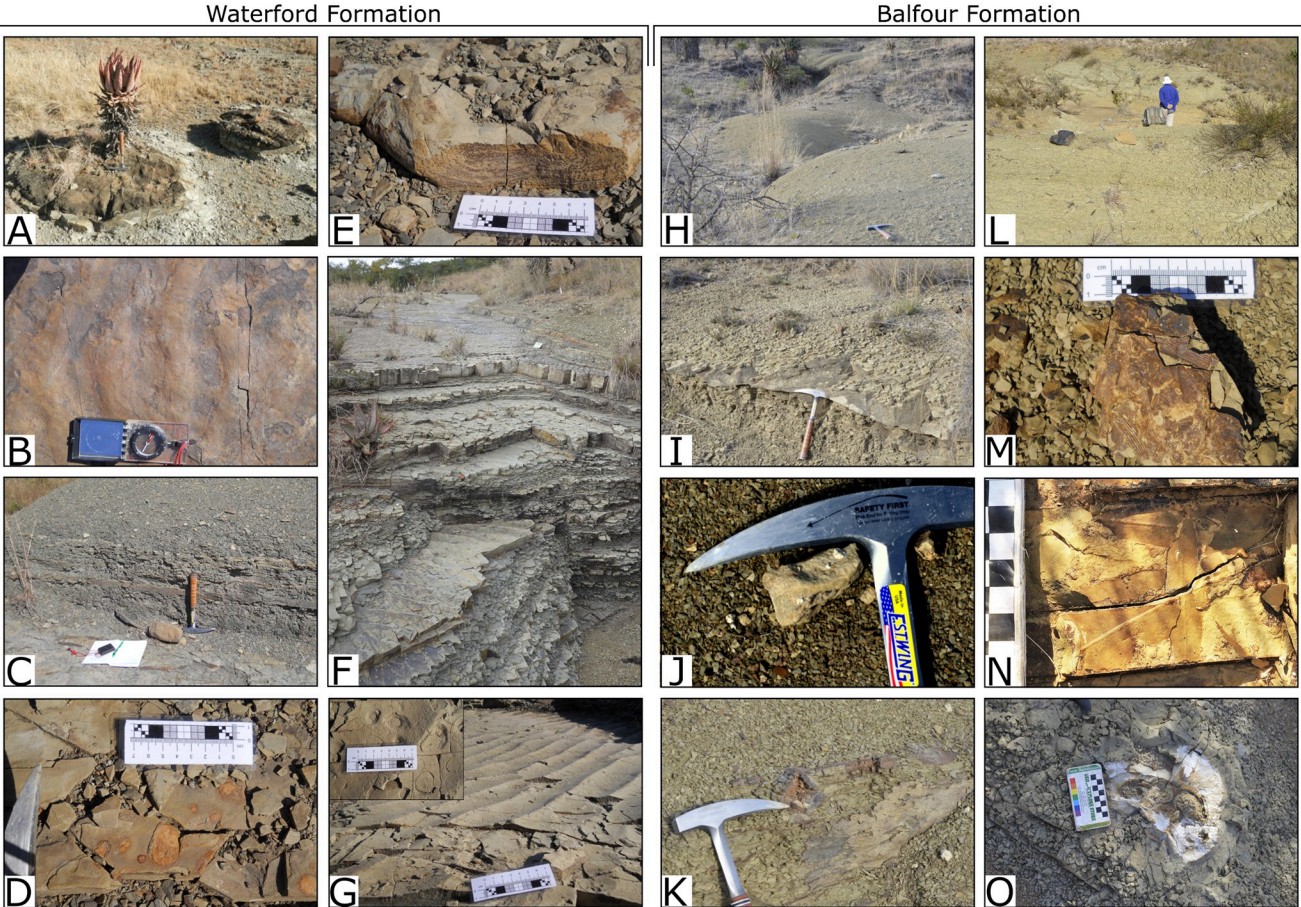

**Fig 3. Features of the upper Waterford and lower Balfour formations in the study area.** A) Large calcareous nodules. B) Rounded/ Smoothed symmetrical ripples on the top of the thick sandstone unit. C) Dark grey, laminated mudstones. D) Iron oxide haloes in sandstone unit. E) Soft sediment deformation. Note how the laminations follow the deformation bed. F) Sequence of rhythmites, comprising alternating mudstone and sandstone units. G) Large symmetrical ripples with sharp, straight crests and *Skolithos* burrows. H) Yellow-green mudstones typical of this facies association. I) Sandstone lens showing lateral/ downstream accretion. J) Calcareous concretions within a sandstone. K) Slickenside surfaces and nodules in mudstones. L) Fossilised wood weathering out of the mudstones. M) Horsetail impressions. N) *Glossopteris* leaf impressions. O) Large, unidentified bone c.f. dicynodont.

sphenophyte stems (*Calmites* and *Paracalamites*; DPG pers. obs.), as well as silicified wood identified as *Agathoxylon africanum* (M. Bamford pers. comm. 2019; Groenewald 2021) (Fig 3). Plant fossils below the palaeosurface are generally more fragmentary than those above it. Vertebrate fossils below the palaeosurface are restricted to isolated and fragmented fish bones and scales, whereas vertebrate fossils recovered from the Balfour Formation by us and previous workers on the farm van der Merwe's Kraal 972 include a partial rhinesuchid amphibian skull (BP/1/7858; c.f. *Laccosaurus*) and fragmentary dicynodont material. Although the body fossils are not diagnostic, the lowermost vertebrate assemblage zone of the Beaufort Group in this part of KwaZulu-Natal is the late Permian (Lopingian) *Daptocephalus* Assemblage Zone (AZ), although it is unclear which subzone is present [37, 57].

## Materials and methods

We recorded the lithostratigraphy and sedimentology of the study area by measuring a stratigraphic section through the exposures of interbedded sandstones, siltstones and mudstones along the stream bed that crosses the palaeosurface. For this we used a Jacob's staff and Abney

**Table 1. Descriptions and interpretations of lithofacies types [modified after Miall [60]; * denotes newly erected lithofacies types from Groenewald *et al.* [37]] from the studied stratigraphic interval.**

| Lithofacies Code | Lithofacies | Description | Interpretation |
|---|---|---|---|
| *St* | Trough cross-bedded sandstone | Medium- to coarse-grained sandstones with cross-bedded units in which the lower bounding surfaces are curved and truncate other facies or cosets. | Migration of three-dimensional dunes. |
| *Sp* | Planar cross-bedded sandstone | Fine- to coarse-grained sandstones in which the foresets dip between 15° and 35°. | Migration of two-dimensional dunes. |
| *Sr* | Ripple cross-laminated sandstones | Very fine- to medium-grained sandstones with ripple cross-laminations. | Lower flow regime, wave and wind induced ripples. |
| *Sm* | Massive or weakly graded sandstone | Very fine- to coarse-grained sandstones in which no internal sedimentary structures are observed. | Rapid deposition, e.g., sediment gravity flows. Alternatively, loss of sedimentary structures could be due to bioturbation and/ or weathering. |
| *Sd** | Sandstone/siltstone beds with deformation structures | Fine-grained sandstone and siltstone beds that display soft-sediment deformation structures including load casts, ball and pillows, and water escape and flame structures. | Rapid sedimentation, liquefaction, reverse density gradation, shear stress, or collapse of channel banks. |
| *R** | Rhythmically interbedded mudstones, siltstones, and sandstones | Alternating heterolithic beds of fine- to medium-, and occasionally coarse-grained sandstones, siltstones, and mudstones. Lower bounding surface of coarser beds is sharp, whereas the upper bounding surface commonly has symmetrical ripple marks. | Deposition in parts of the interdistributary bay proximal to subaqueous distributary channels. Finer-grained argillaceous beds were deposited through suspension settling during periods of lower energy while the interbedded coarser-grained beds were deposited under higher energy conditions. |
| *Fl* | Laminated fines | Relatively thick (> 0.5 m) successions of horizontally and ripple-cross laminated mudstones. | Suspension settling, or deposition under lower flow regime conditions. Depositional settings include distal interdistributary bays, lagoons, and on tidal flats; away from channels and where wave activity is minimal. Fluvial settings include ephemeral floodplain pools and ponds. |
| *Fm* | Massive siltstones and mudstones | Massive beds of siltstone and mudstone displaying no apparent bedding planes. Co-occurs with Facies *Fl* and thin sandstone lenses may be present. | Either through suspension settling, with bioturbation, diagenesis, or weathering resulting in the lack of sedimentary structures, or through rapid deposition e.g., hyperpycnal/ hypopycnal flows. |

level, noting the lithology, colour, sedimentary structures, and fossils present. These data were used to characterise lithofacies and lithofacies associations as well as architectural elements after Miall [58, 59] to provide palaeoenvironmental context to the deposits (Table 1). A recent detailed appraisal of the sedimentology and lithostratigraphy across the Ecca-Beaufort contact in the north of the main Karoo Basin may be found in Groenewald *et al.* [37] and is referred to herein. The described study complied with all relevant regulations and the necessary permit (REF: SAH19/13092) was obtained from KwaZulu-Natal Amafa and Research Institute.

Due to the large area of the palaeosurface and the relatively shallow impressions that are preserved, traditional casting methods were not successful. Consequently, we combine high-resolution surface scanning and aerial images to digitise and accurately record the surface.

An orthophotograph of the palaeosurface was created using 16 photographs taken with a DJI (Shenzhen, China) Spark unmanned aerial vehicle (UAV). The photographs were taken 9.8 m meters above the surface (f/2.6, 1/100s, at 25 mm focal length) using the integrated 1/2.3" CMOS (Effective pixels: 12 MP) sensor and gimble. Each photograph was automatically GPS tagged using the UAV's built in GPS functionality and later stitched into the encompassing orthophotograph using Microsoft Image Composite Editor 2.0.

In addition to photographing the large traces, high-resolution three-dimensional models for six of the better-preserved large traces were produced by scanning using an Artec Eva (Luxembourg City, Luxembourg; http://www.artecthree-dimensional.com/hardware/artec-eva/) handheld structuredlight 3D scanner and processed in Artec Studio 10 and Artec Studio 11 software. The models were exported as Polygon File Format (.ply) and processed further using

the software CloudCompare 2.9.1 (https://www.danielgm.net/cc/) and ParaView v. 5.10.1 (https://www.paraview.org/).

# Results

## Sedimentology

The Dave Green palaeosurface ichnofossils are preserved on the upper bedding plane of a ~10 cm thick ripple-marked fine-grained sandstone bed with a thin (~2 mm) mudstone drape, which we interpret to be situated in the uppermost part of the Waterford Formation (Fig 4).

The exposed palaeosurface is formed in a succession of interbedded mudstones, siltstones, and sandstones in the upper 0.5 m of the Waterford Formation, that are in turn overlain by mudstones and sandstones of the Balfour Formation (Fig 1). Preserved asymmetrical ripples are well defined (Fig 4A and 4B) with sinuous crestlines and well-developed bifurcations, especially in the western half of the surface. The ripples have a wavelength of 35 mm and an amplitude of 3 mm. Palaeocurrent readings from the asymmetrical ripples, with ripple crest strike orientation of 168-348º, indicate a westerly current direction of 258˚. Current-modified and rounded, smooth-topped ripples with rill marks and *Gyrochorte*-like invertebrate traces are present on the northwestern part of the surface (Fig 4C and 4D).

Several medium-sized depressions (>20 cm diameter) contain symmetrical ripples with a strike orientation of 98–278˚, almost perpendicular to that of the asymmetrical ripples (Fig 4E). Large, smooth areas surrounding asymmetrically ripple-marked sections (Fig 4F and 4H) were observed in many parts of the surface; in such areas, no ripples are preserved, or the amplitude of the ripples is much smaller, while the wavelength remains the same. Numerous smaller, subcircular depressions, many with asymmetrical ripples, are also preserved (Fig 4F and 4G). We consider these to represent trackways with poor morphological preservation and describe them further below under Subcircular depressions.

## Ichnology

The exposed palaeosurface is covered with numerous traces and tracks of various size and form (Fig 5). For the purposes of this paper, we will describe and discuss the different tracks in three groups based on the morphology and arrangement of the traces: "Large impressions", "Subcircular depressions", and "Linear traces".

**Large impressions.** At least seven large elongate and spindle-shaped impressions are preserved on the surface (Figs 2, 5 and 6; Table 2). These impressions vary in length between approximately 1.5–2.1 m, and the shape of each trace may be described in three sections, namely: 1) a tapered "tail" opening towards, 2) an expanded intermediate region (present in impressions 3, 4, 5), and 3) a wide rounded rectangular "body" between 200 mm and 220 mm wide (Fig 6). The outer edge of the impressions manifest as pushed-up ridges or bulges. Some of the impressions are followed or preceded by smoothed traces 120–190 mm wide and up to several metres long, which generally interrupt or obliterate the ripples. These smooth traces connect some of the impressions (e.g., Impressions 3, 4, 5) and reveal two large semi-circular patterns (Fig 5). Vague sinusoidal patterns are present in some of the smooth traces e.g., Impressions 3 and 5.

Impression 2 is 3.15 m long and consists of two superimposed impressions, with the tail section of the second impression (B) overprinting on the body section of the first (A; Fig 7). The length of Impression 2A is ~2.09 m, whereas that of Impression 2B is ~1.52 m. The width for the body portion of the impressions varies between ~180 and 200 mm, with the expanded section in Impression 2B attaining a width of ~230 mm. A low ridge is present along the midline of Impression 2 (Fig 7).

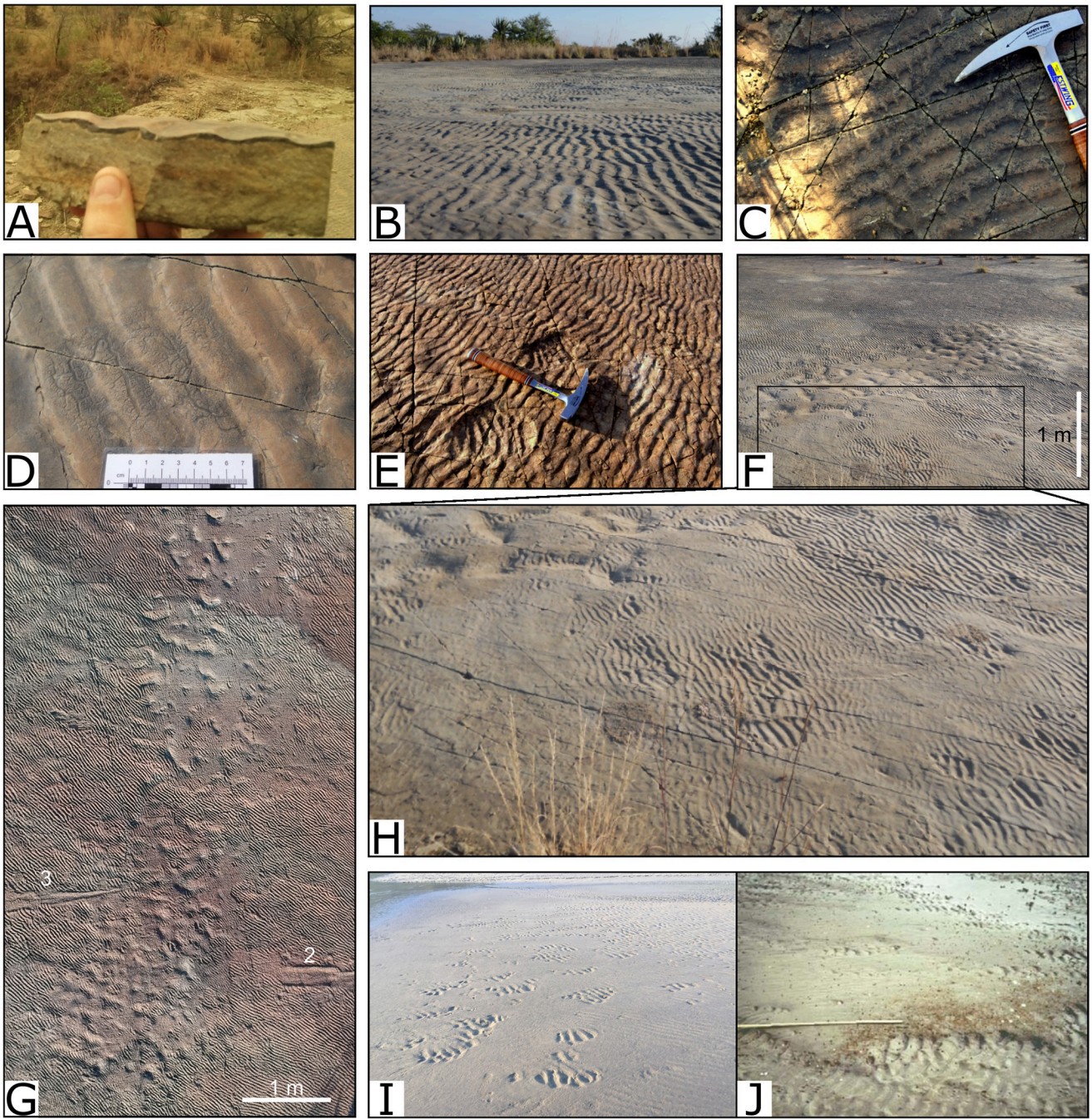

**Fig 4. Sedimentary structures preserved on the palaeosurface.** A and B) Well preserved asymmetrical ripples indicate a westerly current direction. C) Current-modified ripples, with tongues of sand on the lee side of the ripples, in the northwestern part of the surface. D) Rill marks on smoothed ripple marks in the northwestern part of the surface. E) Depressions with interference ripples F) Smooth, unrippled section in the foreground with rippled surface in the background and a 'corridor' of subcircular to blob-shaped depressions (tracks) between the two. G) Aerial view of the portion of the 'corridor' passing between impressions 2 and 3. H) Closer view of smooth areas surrounded by ripple marked depressions which could indicate the presence of a microbial mat on the surface, as seen in modern environments (I and J). I) Modern example of smooth areas surrounded by ripple marked depressions at the Mbotyi River Mouth, South Africa. J) Modern example of an algal mat and ripple patches from Mellum Island tidal flats, Germany (modified from [61]).

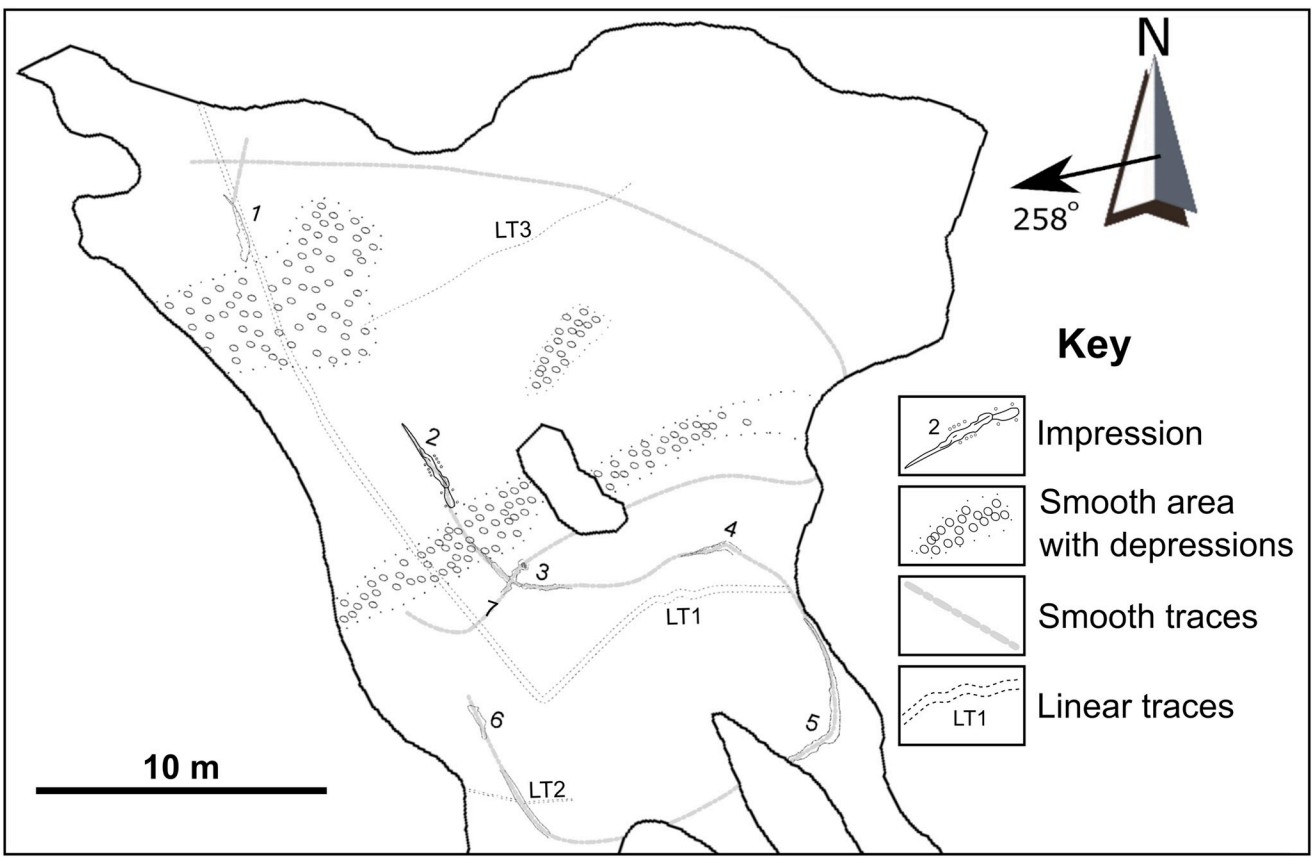

**Fig 5. Sketch map of the palaeosurface showing the location of the seven large impressions, smooth areas with subcircular depressions, and linear traces.** The palaeocurrent as indicated by the asymmetrical ripples is 258˚. The six large impressions are numbered 1–7 and this scheme is followed in the text and subsequent figures. The linear traces are labeled LT1 –LT3.

At least 15 round depressions, interpreted as footprints, are preserved between 90 and 160 mm away from the body impressions in Impression 2 (Fig 7). These depressions, eleven of which are alongside impression 2A and four are alongside impression 2B, have diameters between 33 and 63 mm and some have smooth expulsion rims. The footprints have poor morphological preservation (M-preservation grade 0 using the scale of Marchetti et al. [62]) that makes it difficult to distinguish manus from pes. However, there are only four footprints adjacent to impression 2B, with the posterior pair of footprints positioned between 530 and 590 mm from the anterior pair for the right and left, respectively (RP3-RM5 and LP3-LM4). A similar distance separates the supposed posterior and anterior pairs of footprints, taking into account progression while generating the body trace (LP1/RP1-LM2/RM2 or LP2/RP2-LM3/RM4), in impression 2A (Fig 7). This most probably corresponds to the gleno-acetabular distance, which is reported to be a good estimate of the body length of a tracemaker [63–65]. As such, we estimate the body length of the tracemaker to be approximately 560 mm.

Impression 3 (Fig 6 and S1 Fig) has a length of ~1.59 m and a body width of between 180 and 210 mm, pinching down to ~140 mm at the juncture between the body and expanded intermediate section. A low ridge is present in the centre of the expanded intermediate section and near the point where the tail and intermediate portions join. Impression 3 leads into a 120–130 mm wide smooth trace (S1 Fig) and is cut by a second smooth trace that is 120–145

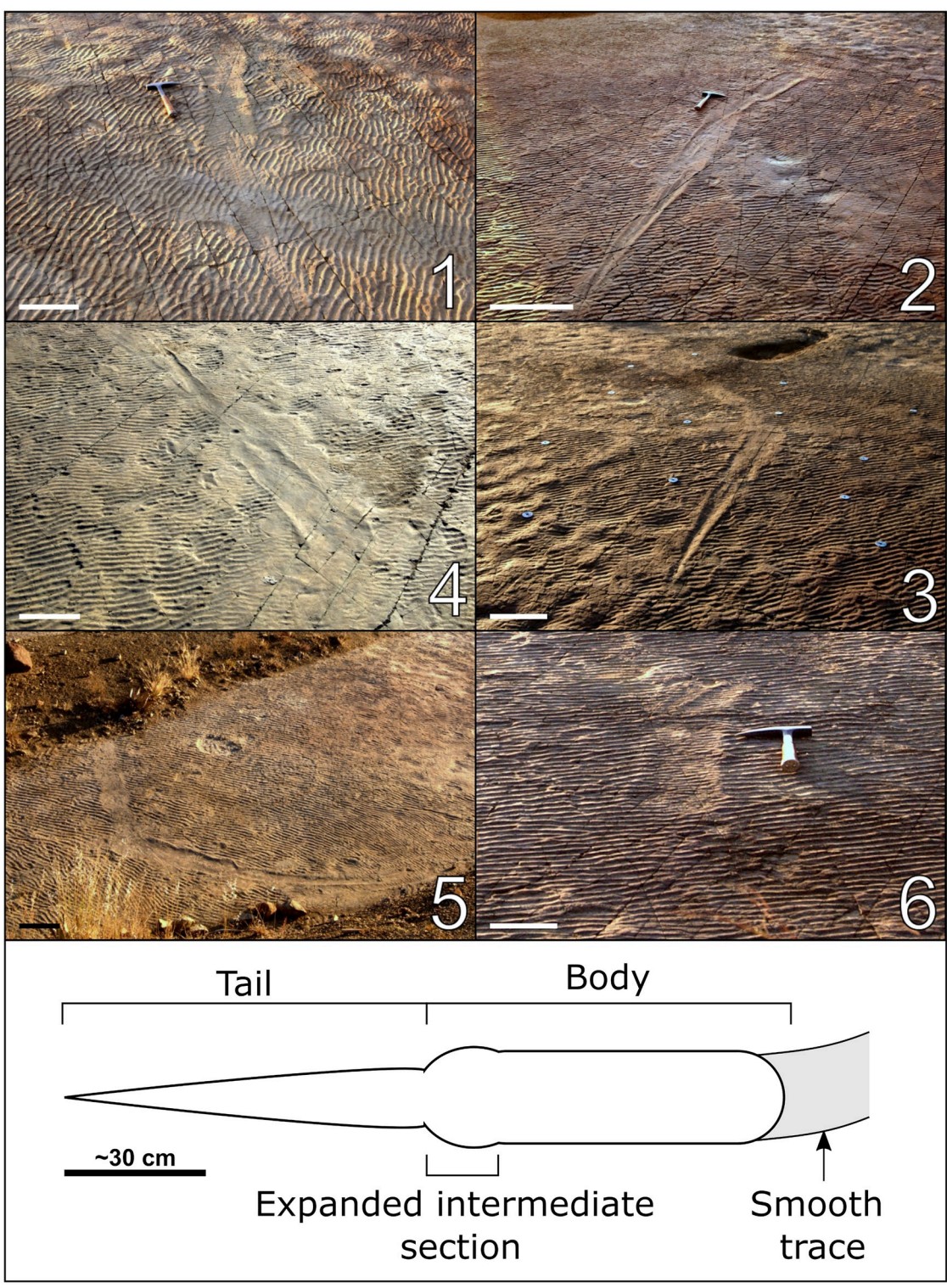

**Fig 6. Oblique field photographs of the six best preserved large impressions from the Dave Green palaeosurface and simplified sketch of an impression showing the characteristic "body", "tail", and intermediate expanded sections.** Numbering of the impressions corresponds with the rest of the figures and text. In parts 1–6: scale bars equal 30 cm.

**Table 2. Dimensions of the better-preserved large impressions and selected smooth traces.**

| Trace | Length (cm) | Width (cm) |
|---|---|---|
| Impression 2A | 209 | 18.0–20.0 |
| Impression 2B | 152 | 18.0–23.0 |
| Impression 3 | 159 | 18.0–21.0 |
| Impression 4 | 161 | 15.0–18.0 |
| Impression 5 | 159 | 18.0–21.0 |
| Smooth trace following impression 3 | - | 12.0–13.0 |
| Smooth trace following impression 4 | - | 19.5–20.0 |
| Smooth trace following impression 5 | - | 13.5–18.5 |
| Smooth trace following impression 7 | - | 12.0–14.5 |

mm wide and which crosses in front of impression 3. A faint impression (impression 7) is present near the inferred start of this second smooth trace (S1 Fig).

Impression 4 (Fig 6 and S2 Fig) is 1.61 m long, and the width of the body portion varies between 150 and 180 mm. It is also followed by a ~195–200 mm wide smoothed trace.

Impression 5 (Fig 6 and S3 Fig) is ~1.59 m long and 180–210 mm wide in the body section, narrowing to about 160 mm at the point between the body and intermediate portions. A small, ovoid depression (~38 mm wide, ~60 mm long), possibly a footprint, is present 125 mm to the right and near the back of the body impression (S3 Fig). A low ridge is present along the midline of the impression and is most prominent around the juncture between the tail and intermediate portions. Impression 5 is followed by a wavy trace 135–185 mm wide (Fig 6 and S3 Fig).

**Subcircular depressions.** Several groupings of smaller, subcircular and blob-shaped depressions occur across the palaeosurface. Three such groupings, indicated in Fig 5 as "Smooth area with depressions" since the area surrounding the depressions is often smooth and not rippled, are: 1) a "corridor" ~0.9 m wide that crosses the surface from the western to eastern side (Fig 4G and Fig 8A–8E); 2) a concentration just north of the present-day island; and 3) a higher density of these depressions preserved in the northwestern part of the surface (Fig 8). The depressions vary in size and shape with diameters ranging from 10–15 cm. The bottom of many of the depressions is sculpted with asymmetrical ripples and little-to-no morphological details are preserved.

**Linear traces.** Several examples of single or paired linear traces are preserved on the palaeosurface. These traces are not very deep and are most readily observed during the low-angle light of early morning. Three of these, LT1, LT2, and LT3, are indicated on the map (Fig 5).

The first linear trace (LT1) consists of a paired trace that can be followed southwards from the northwestern edge of the surface, where it either overprints or is overprinted by Impression 1 (Fig 9A and 9B). It makes a sharp turn just south of Impression 7 (Fig 9C and 9D) and can be followed eastwards across the surface to the margin of the palaeosurface (Fig 9E). The individual traces are ~1 cm wide and the paired traces are ~35 cm apart. Except for a short, sinuous section near Impression 4, the trails are relatively straight. In the sinuous section (Fig 9F), the trails have a wavelength of 36–39 cm and an amplitude of 5 cm.

A second linear trace (LT2) is present in the southeastern corner of the surface. LT2 consists of an approximately 7 cm wide paired straight trace that can be followed for roughly 4 m and crosses the smooth trace connecting Impressions 5 and 6 (Fig 9G). LT2 affects the tops of the ripple crests but not the troughs.

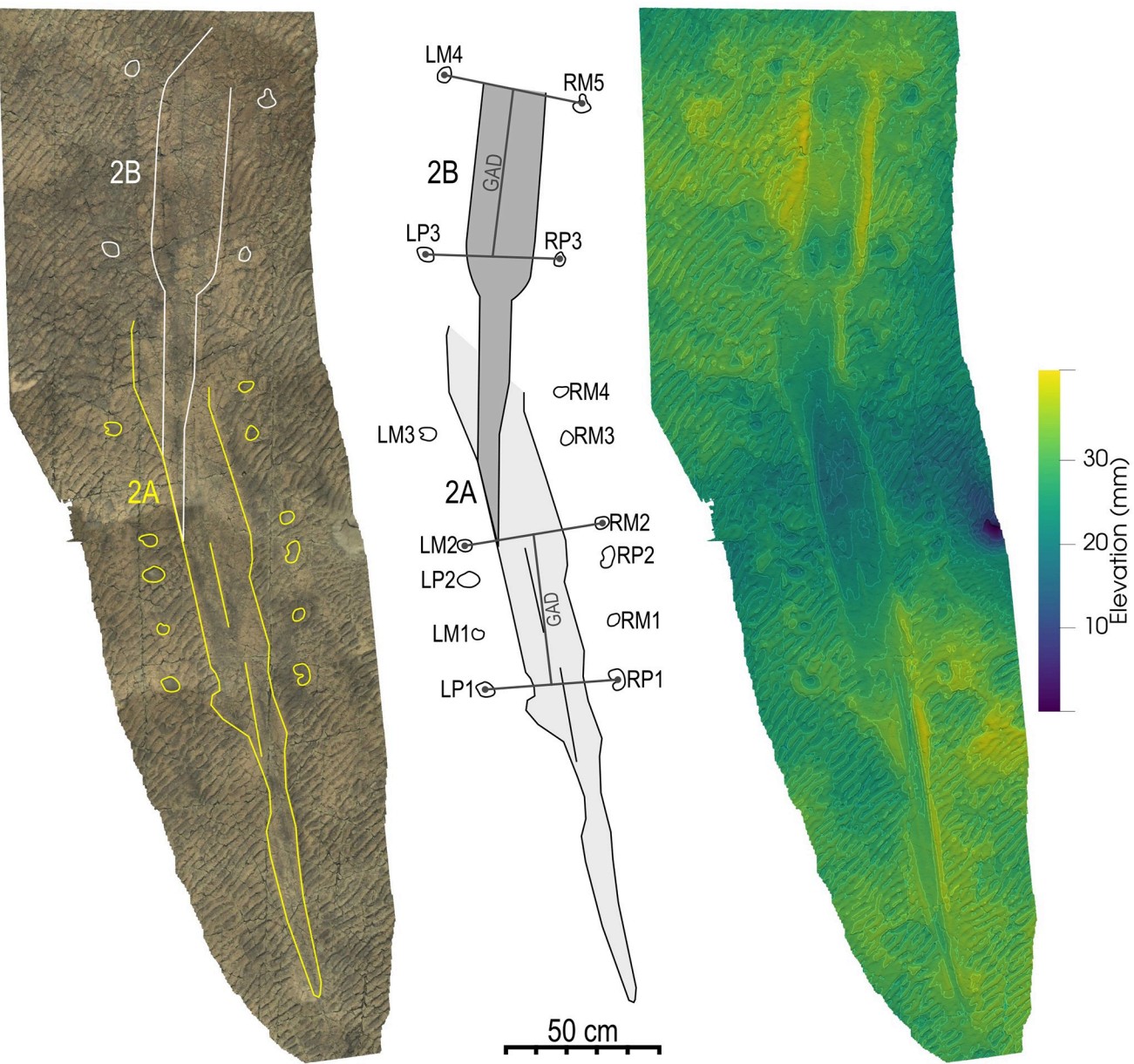

**Fig 7. Details of Impression 2.** Textured scan (Left) with the outline of the impressions [2A (yellow) and 2B (white)] and footprints compared to the false-colour depth model (Right). Depth scale is in mm.

A third linear trace (LT3) cuts across the northern portion of the surface. Although faint, this long unpaired trace is almost 8 m long and disrupts both the ripple crests and troughs (Fig 9H).

Northwest of the mapped palaeosurface, a further two linear traces were observed on the rippled sandstone surface. The first of these is a ~3 mm wide sinuous trace with a wavelength of ~3 cm and amplitude of ~2 cm (Fig 9I). The size of the trace allows for it to be attributed to *Undichna unisulca* and not *Cochlichnus*, as differentiated by Minter and Brady (2006). The second is a straight paired trace, with the two ~3 mm wide traces 3 cm apart (Fig 9J). This trace disturbs the ripple crests and some of the troughs.

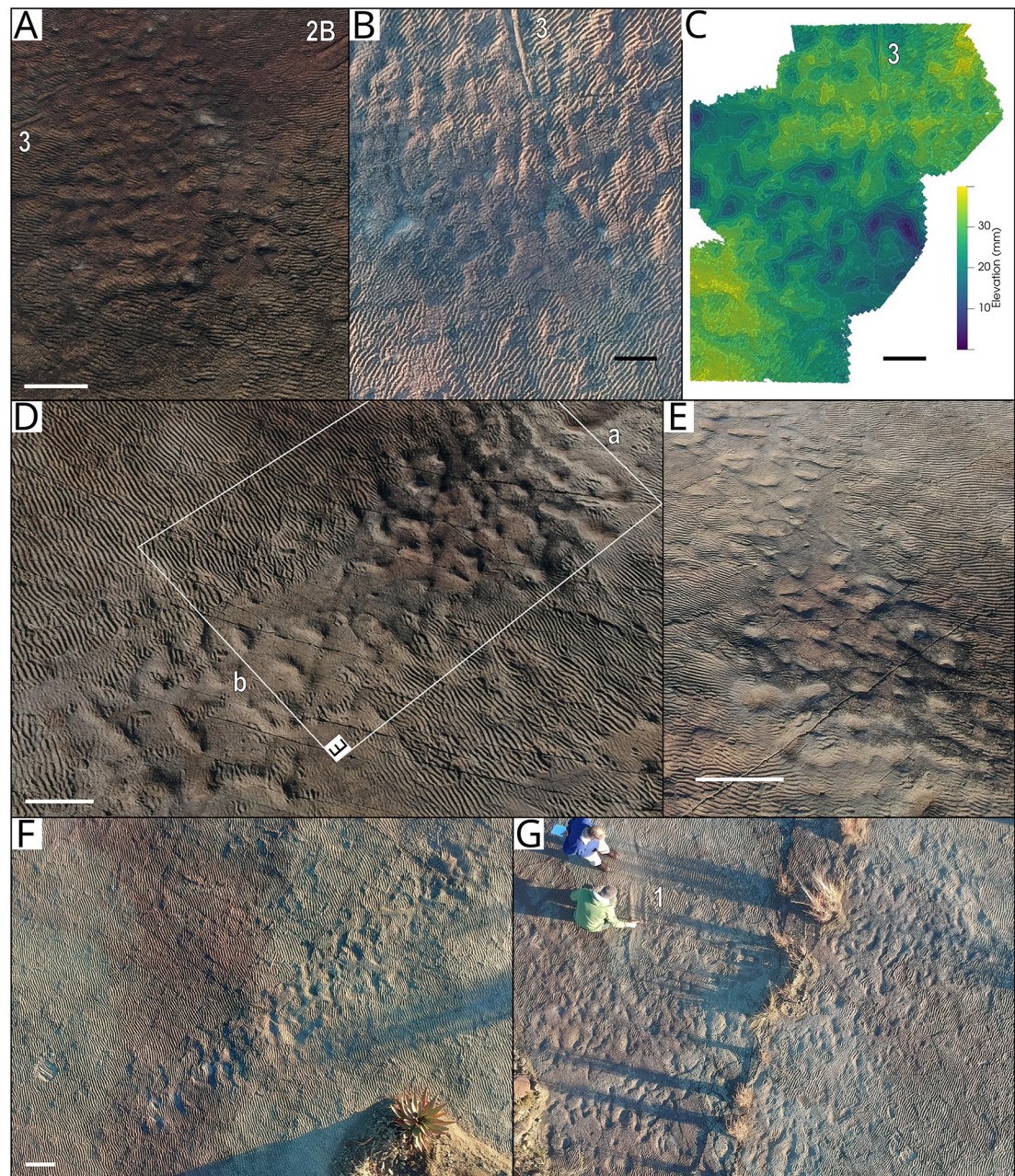

**Fig 8. Groupings of subcircular depressions interpreted as footprints on the Dave Green palaeosurface.** A) Aerial oblique photograph of the western portion of the "corridor" of footprints between Impression 2 and 3. B) Overhead photograph of the 'corridor' between Impressions 2 and 3 with the false-colour depth model (C). D) Overhead view of the eastern portion of the 'corridor' with an oblique view of the footprints between points 'a' and 'b' (E). F) Overhead view of the concentration of depressions just north of the present-day 'island'. G) Overhead view of the depressions in the northwestern part of the surface. Large impressions are numbered in A-C and G. Scale bars equal 30 cm (A, D-F) and 50 cm (B and C).

## Discussion

### Depositional setting

Based on sedimentological field observations, we interpret the Dave Green palaeosurface to represent a sandy tidal flat or the sandy floor of a shallow embayment or lagoon. The traces

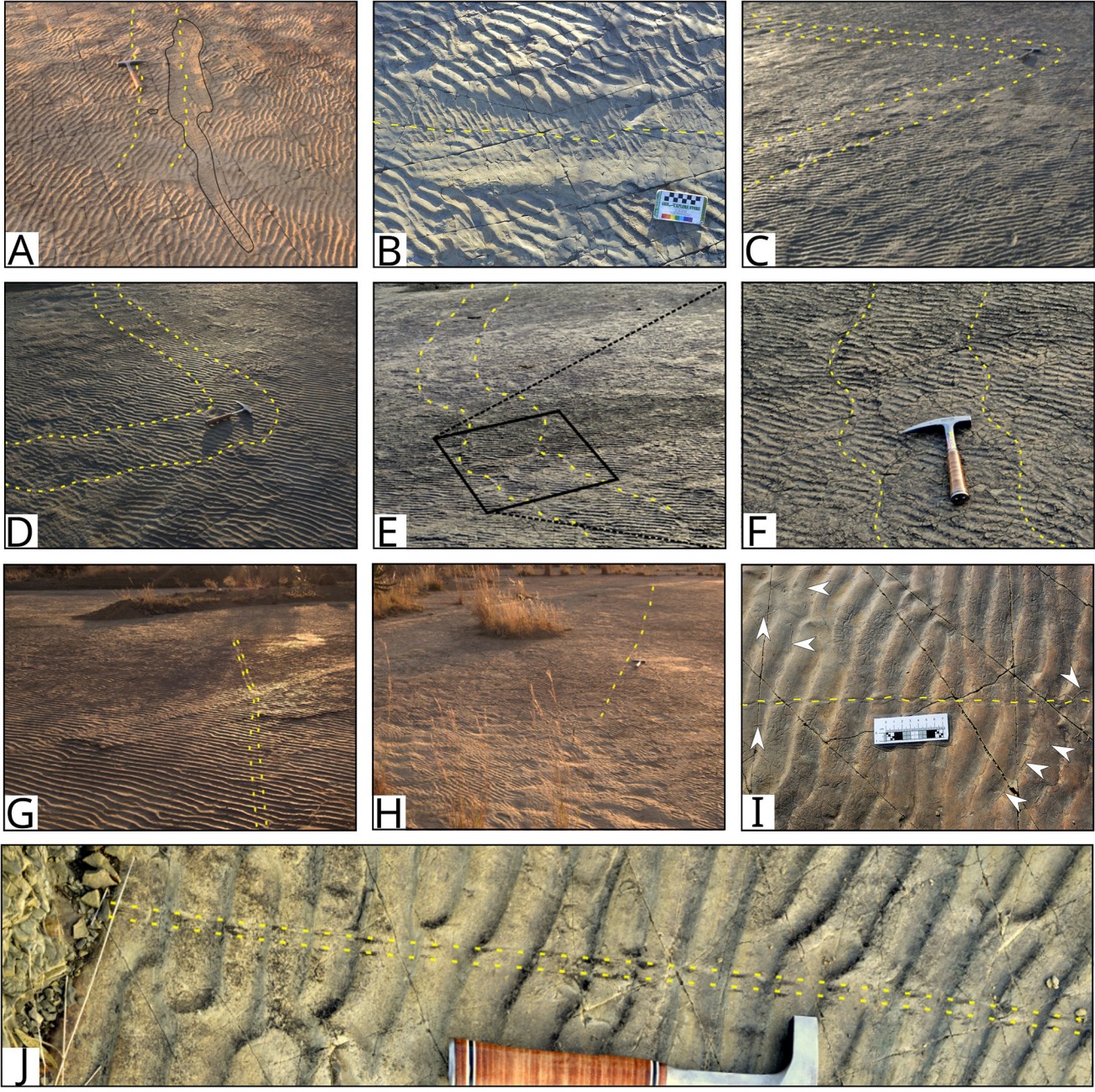

**Fig 9. Linear traces preserved on the palaeosurface.** A-E) A large, paired trace (LT1) can be followed southwards from the northwestern edge of the surface, overprinting (or overprinted by) Impression 1 (A and B). The trace makes a sharp turn (C and D) and heads eastwards (E). The trace is mostly straight except for a small sinuous section (F). The box outlined in E is the area shown in F. G) Paired trace (LT2) cutting through ripple crests only. The trace crosses the smoothed swim trail in the Southwestern corner of the surface. H) Long linear trace (LT3) on northern part of the surface. I) *Undichna* trace in north-western part of the surface (Off mapped area). Arrows show unidentified invertebrate traces. J) Linear paired trace cutting ripple crests in north-western part of the surface (Off mapped area). Hammer (~30 cm) for scale (A, C, D, F, H and J). Scale card in B = 8 cm and numbers on scale card in I = cm.

made by fish, which overprint some of the large impressions and are therefore younger, suggests that the palaeosurface was submerged at the time that the large impressions were made and *Gyrochorte*-like traces in the northwestern part of the surface provide evidence for a moderate energy, near-shore to shallow water environment [66]. The water was likely

brackish-to-fresh based on the conclusions of several studies using trace elements from the upper Ecca Group [51, 67, 68]. The presence of unrounded asymmetric ripple crests, with the exception of some ripples in the northwestern part of the exposure, suggest that the surface was not reworked and rill marks on the surface of some of the smoothed ripples (Fig 4D) indicate the subsequent fall in water level leading to subaerial exposure of at least the ripple crests [69–71]. However, that the surface did not dry out completely before being buried by mud is suggested by the absence of desiccation cracks or other evidence for extended exposure.

Depressions on the mapped surface that contain symmetrical ripples (Fig 4E) with crest strike orientations differing from that of the asymmetrical ripples by around 70˚, could have formed as erosional depressions, or possibly footprints, which remained filled with water following the drop in water level. Wind blowing across the surface of these water-filled depressions could have resulted in the formation of the symmetrical ripple marks.

The smoother, non-rippled areas of the surface are possibly where a biofilm or microbial mat was growing on the sand in a shallow water environment and similar structures can be seen on present day tidal flats (Fig 4I and 4J) [61, 69, 70, 72–74]. The transition between the rippled and non-ripples areas is smooth, indicating that the surfaces are penecontemporaneous [74]. In parts of the palaeosurface, the surface weathers with distinctive quadrangular chips. These resemble 'mat chips' produced by the erosion of modern microbial mats by waves and currents [72] and further support the presence of a microbial mat on the surface. Tracks and traces require the substrate to be 'just right' in order to be preserved [75]. Under certain conditions, the presence of microbial mats have been shown to favour or enhance the preservation of footprints [76, 77] and microbial mats are commonly associated with fossil trackways, including in the main Karoo Basin [2, 9, 25, 29]. In the case of the Dave Green Palaeosurface, microbial mats likely stabilised the surface sediments, preventing the formation of ripple marks in the smooth areas, and could have also influenced the mineralisation of the drape on the surface [61, 73].

## Potential tracemakers

**Large impressions.**   The seven large impressions preserved on the palaeosurface are interpreted to be body and tail impressions formed as temporary resting traces [cubichnia; 78], whereas the smoothed linear traces are interpreted as locomotory (swim) traces [repichnia; 78]. Superficially, the large impressions preserved on the palaeosurface are similar in appearance to some present-day examples produced by crocodiles and alligators (Fig 10B) [79, 80]. However, crocodylomorphs do not appear in the fossil record until the Upper Triassic/ Lower Jurassic [81–87] and the earliest crown-group crocodilians are from the Late Cretaceous, or possibly a little older [86, 88, 89]. Since the Dave Green ichnofossils are late Permian in age, it is highly improbable that the traces were made by a crocodylomorph. The only plausible candidate for the maker of the large impressions and traces is a rhinesuchid temnospondyl (Fig 10C), since this clade is the only group of tetrapods with a "crocodile-like" body plan known from the *Daptocephalus* AZ, and indeed the immediate vicinity of the study area [57, 90–92].

Based on the preserved impressions, the tail and body of the tracemaker was ~ 1.59 m long. Adding a skull with a length of ~30 cm, typical for late Permian rhinesuchids, gives a total length of 1.89 m for the tracemaker. This is within the size of *Uranocentrodon senekalensis*, which has a skull length of up to 50 cm and a total adult body length of between 2.3 and 3.75 m [94]. *Laccosaurus watsoni* is slightly smaller with a skull length of approximately 23 cm for the holotype [95] but too few specimens are reliably attributed to this genus to estimate it's maximum body size.

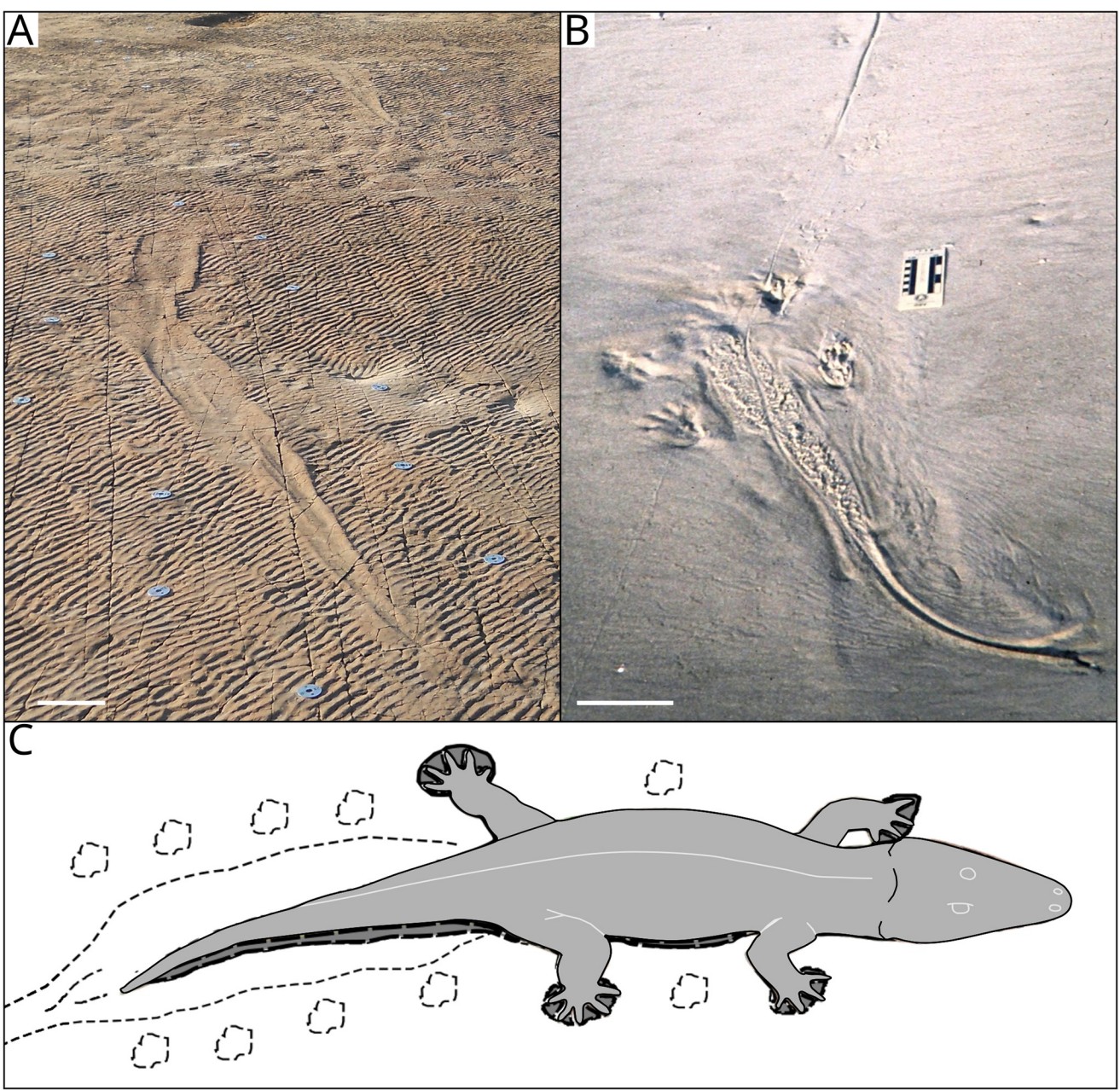

**Fig 10.** Comparison of Impression 2 from the palaeosurface (A) with a present-day body impression and trail of *Alligator mississippiensis* on the foreshore at St. Catherines Island, Georgia, United States (B). Photograph courtesy of St. Catherines Island Sea Turtle Program, Gale A. Bishop and modified with permission from [93]. C) A rhinesuchid temnospondyl such as *Laccosaurus* or *Uranocentrodon* is probably the tracemaker. Scale bar = 30 cm (A and B).

Body impressions of amphibians showing most or all of the body outline are rare, with only three named ichnospecies described from the Palaeozoic, *Hermundurichnus* fornicatus [96, 97] and *Sauropleura longicaudata* [98, 99] from the Carboniferous-Permian of Europe and *Temnocorpichnus isaacleai* from the late Mississippian of eastern Pennsylvania in North America [100–103]. Amphibian body resting traces associated with trackways and swimming traces have previously been described by Turek (1989), but were not named. All these examples are, however, as much as 10 times smaller than the impressions reported here.

While some species of stereospondyls, including rhinesuchids such as *Uranocentrodon sene-kalensis*, had ventral osteoderms that were likely covered by a horny keratin layer [104, 105], the studied impressions generally have smooth bottoms and do not show any definitive evidence for the tracemaker having ventral osteoderms. Consistency or grain size of the substrate [106], or movement of the animal, smearing the trace and thereby destroying the impressions, could account for the lack of fine morphological detail in the traces. The lack of morphology in the associated footprints could also be the result of the interplay between the algal mat and substrate properties [76, 77]. Alternatively, post-registration growth of the microbial mat on the substrate could have resulted in the loss of morphological detail over time [61, 69, 70, 73, 74, 76, 77, 107].

**Subcircular depressions.** We interpret the numerous smaller, subcircular to blob-shaped depressions on the surface as likely representing footprints made by a group or several groups of medium-large terrestrial tetrapods that waded and crossed through the water, rather than being erosional depressions. This is based on the grouping of the depressions as well as the 'corridor' that crosses the surface. The lack of morphology in the depressions is an indication of the fluid, liquified character of the substrate [106, 108]. The thin upper layer of the palaeosurface comprises a finer grained, mineralised drape that was capable of preserving smaller traces such as fish fin traces and *Cochlichnus* traces, but the walls of deeper depressions penetrating the more liquefied underlying substrate collapsed. The act of larger animals walking on the substrate would have torn the microbial film, exposing the sand underneath while the current was strong enough to form ripples in the base of these depressions. This indicates that these trackways were probably made before the temnospondyl traces. The morphology of the traces provides little information on the identity of the trackmaker but the most likely candidates are dicynodont therapsids. Trackways made by groups of dicynodonts have been reported previously from the Karoo [12, 13], and they are the most common animals of the required body size recorded by body fossils from the area.

**Linear traces.** The ichnotaxon *Undichna* is used to describe traces that are sinusoidal or scalloped, single, paired, or multiple sets of overlapping grooves that form regular, repeating patterns in bottom sediments of aquatic environments [1, 78, 109, 110]. These patterns are thought to have been made by pectoral, pelvic, anal, and caudal fins of fish dragging along the substrate during swimming behaviour [1, 78, 110, 111]. Differences in the amplitude and wavelength of these patterns are related to the fish size and shape, direction and velocity of the water current, and swimming speed [110]. As such, we interpret the linear and sinusoidal traces seen on the palaeosurface as likely being created by fish of various shapes, sizes, and swimming styles. While most of the traces are straight and lack the sinusoidal pattern characteristic of *Undichna*, this could be explained by the fish drifting with the current, or swimming with thunniform or ostraciform motion, resulting in the much lower amplitude of the pectoral fins and, thereby, of the trace itself. The sharp turn seen in LT1 could have been avoidance behaviour of the fish. It is also possible, however, that some of the more linear examples such as LT2 and LT3, which have a similar orientation to the palaeocurrent direction, were created by a stick drifting and dragging along the sediment, thereby forming a pseudo trace fossil (R. Gess, pers. comm. 2015).

## Rhinesuchid locomotion

The trackway preserved on the Dave Green palaeosurface provides evidence for both swimming and bottom walking behaviours. Because of this, and in contrast to many present-day and fossil examples of crocodilian and temnospondyl trackways, which typically comprise a trackway with a central tail or belly drag [28, 29, 79, 80], the impressions and traces on the

Dave Green palaeosurface lack associated footprints or scratch marks, with the exception of Impression 2.

Temnospondyl locomotion has traditionally been compared with extant examples of salamanders, newts, and crocodilians, since these are considered to have similar bauplans–sprawling gait with limbs of equal size, elongated trunk, and a tail–and are capable of both aquatic and terrestrial locomotion [29, 112–114]. Work by Marchetti [115] and Marsicano et al. [29] has suggested that large temnospondyls used mainly their forelimbs rather than hindlimbs for propulsion during terrestrial locomotion, as a consequence of their relatively larger skull, heavier pectoral girdle, and relatively reduced pelvic girdle and tail.

When considering the locomotory capabilities of stereospondyls, a number of different categories, ranging from terrestrial crawlers to fully aquatic swimmers, are recognized [105]. With the exception of some terrestrially adapted species e.g. *Lydekkerina huxleyi* [116, 117], most stereospondyls have small limbs with rudimentarily ossified humeri, femora and girdle elements. This suggests limited terrestrial locomotory ability and most stereospondyls are therefore thought to have been aquatic [105, 117], an interpretation further supported by histological studies [118]. Some forms were adept swimmers, with a long, laterally compressed tail probably providing propulsion and the gracile forelimbs being used more for manoeuvring than active swimming [105] or when performing bottom-walking behaviour in a manner similar to extant crocodiles [114, 119]. Others resembled the modern cryptobranchid salamanders in body proportions and were likely less active swimmers but sufficiently agile to lunge and capture larger prey items [105, 120].

Impressions 2 and 3 are relatively close together and appear to follow on from one another (Figs 2 and 5). As such, we consider these to have been formed by the same individual. The direction of movement can be determined by looking at the tail to body direction. For example, Impression 2 indicates an animal that was moving in a south-easterly direction, as evinced by the tail thickening in the body direction as well as the more pronounced expulsion rim along the posterior margin of some of the footprints, indicating the direction that the animal pushed against the substrate.

The fact that many of the impressions are isolated and have no connecting trackways or swim traces (e.g., the gap between Impressions 2 and 3) is interpreted as evidence for subaqueous activity [111] i.e., representing where the animal left the ground and either floated or swam for a short distance within the water column before making contact with the substrate again. An alternative interpretation is that the gaps resulted from a preservational bias due to different mechanical properties of the substrate in different areas at the time of registration. However, because of the lack of disturbance or deformation of surface between impressions, especially to the ripple crests, this is considered less likely. Impression 2 (Fig 7), with associated footprints, would have probably been created when the animal touched down and performed bottom walking behaviour. The absence of morphological detail in the footprints is likely because the sediment was under water and very soft [108], and can also be associated with the way that amphibians and other animals walk underwater, since their legs are not bearing the full weight of the body [111, 112, 114, 119, 121, 122]. It should be noted that variable footfall patterns and stride lengths have been reported for underwater walking, depending on the conditions and behaviour, e.g. similar but more variable stride length have been recorded in California newts walking underwater compared to on land [112], whereas much greater paces and strides have been recorded for bottom walking crocodiles, especially when punting or carried by waves [119]. Changes in the footprint trackway associated with impressions 2A and 2B could therefore reflect a shift from bottom walking, with a similar but more variable stride length to terrestrial pedestrianism in 2A, after which the trackmaker pushed up and floated a short distance before settling on the bottom–hence only leaving four footprints associated with

Impression 2B. It is also possible that both impressions were created through walking but that some of the tracks were just not preserved.

Impressions 3 and 5 provide insight into the swimming locomotion of the temnospondyl. Upon entering the water crocodylomorphs progress with lateral undulations of the tail and body [123, 124], tucking their limbs next to the body to reduce drag [123]. A similar swimming motion can be seen in extant salamanders [121, 125]. The expanded area present anterior to the tail in some of the impressions is interpreted as the area where the hind limbs of the trace-maker were tucked in next to the body while swimming. The swim trace preserves slight sinusoidal movements, which are interpreted as having been formed through continuous sub-undulatory propulsion (undulations that have no recovery stroke) in a manner similar to that observed in extant crocodiles [123, 124] and salamanders [121, 125].

Impression 3 shows one trace that is crossed and overprinted by a second trace (from Impression 7) moving from west to east and which leads into a second large circular route that crosses the northern half of the palaeosurface before curving back towards Impression 1 (Fig 5). The large, semi-circular routes of the traces across the surface, and the associated orientation of the animal relative to the current along the route, with many impressions facing into or perpendicular to the inferred palaeocurrent direction, suggest that the animal was capable of swimming independently of the current. However, the absence of well-defined ripples in most of the body impressions in conjunction with the swim traces indicates that the current was no longer strong enough to overprint the traces with new ripple marks before the surface was buried and preserved by the overlying strata. Microbial mat growth could have also had an influence, especially with Impressions 2, 3 and 4 which occur closer to the smooth areas where inferred microbial mat growth is most abundant, helping solidify the substrate and inhibiting the formation of ripple marks. The lack of ripple marks could also suggest that the water was too shallow or absent at the time registration, although, if this were the case, we would expect to see more footprints associated with the traces.

Some of the traces preserve sharp turns or changes in direction, e.g., Impressions 3, 4, and 5. No observable footprint impressions or disturbances to the adjacent rippled surface are associated with the turns as observed in other amphibian swimming traces [121], and the slightly enlarged area anterior to the tail implies that the hind limbs were tucked against the body. It is possible that the animal used its front legs to help steer in a manner similar to younger crocodiles [124] and that they did not touch the substrate. Another possibility is that the animal did not use its legs at all while swimming, as seen in larger crocodiles [124], but they were used only when the animal touched the substrate, or while bottom walking [116, 119]. While a direction of travel cannot be determined for LT1, it is worth noting that Impressions 1–4 run almost parallel to LT1 (Fig 5). If LT1 is taken to have been travelling north-south before turning and heading west-east, this could suggest that the tracemaker was actively following the fish, and that the traces record some hunting behaviour. Furthermore, the point where the smooth trace and LT1 are closest together, is near the sinuous section, i.e., at which point the fish put in a burst of speed to escape.

Although the presence of multiple body impressions on a single bedding plane in other localities has been interpreted as representing some type of gregarious, possible even mating behaviour–as suggested for *Temnocorpichnus isaacleai* [100], we consider the impressions to represent the behaviour of one, maybe two individuals. This interpretation is based on the fact that many of the impressions appear to follow on from one another and are sometimes linked by the smoothed swim traces to form two circular routes. The second circular route overprints the first one at impression 3 and they were therefore made at different times. However, the lack of well-defined body impressions along the second route limits comparison of body

measurements of the tracemakers and it is uncertain whether they were made by the same individual or by two separate individuals living in the same area.

## Conclusions

The remarkable Dave Green palaeosurface preserves a number of unique trace fossils attributed to medium to large tetrapods, fish, and invertebrates that inhabited a predominantly sandy tidal flat or shallow sandy lagoon in the late Permian. The application and benefit of utilising digital recording methodologies, including 3D surface scanning and UAV photography of palaeosurfaces, is evident here. Among the many trace fossils are at least seven morphologically unique body impressions with associated swim traces, which we interpret as having been made by a medium to large rhinesuchid temnospondyl with a total length of approximately 1.89 m. These traces provide evidence for an active lifestyle of swimming and bottom-walking behaviour in rhinesuchid amphibians during the Changhsingian, possibly while looking for food or hunting. The preserved swim traces also support interpretations that rhinesuchid temnospondyls swam using an undulatory motion of the tail and held their legs tucked in next to the body in a manner similar to extant crocodiles.

## Supporting information

**S1 Fig. Details of impressions 3 and 7.** Textured scan (top), with the outlines of the impressions and associated swim trails of impression 3 (White) and impression 7 (Yellow), shown next to the false-colour depth model (bottom). Depth scale is in mm.
(TIF)

**S2 Fig. Details of impression 4.** Textured scan (Left) with the outline of the impression and swim trace shown next to the false-colour depth model (Right). Depth scale is in mm.
(TIF)

**S3 Fig. Details of impression 5.** Textured scan (Left) with the outline of the impression and swim trace shown next to the false-colour depth model (Right). Depth scale is in mm.
(TIF)

## Acknowledgments

Thanks goes to Dr Natasha Barbolini and Dr Gideon Groenewald for assistance in the field and helpful discussions, and to Dave and Sandra Green for access to the site, assistance, and for providing accommodation during the field visits. Thanks also go to Dr Billy de Klerk, Dr Rose Prevec and Dr Rob Gess for their valuable inputs. We are grateful to Academic Editor Prof. Jörg Fröbisch for handling our manuscript, and to Dr Lorenzo Marchetti and Dr Eudald Mujal for their constructive reviews. Opinions expressed and conclusions arrived at, are those of the authors only and do not necessarily reflect those of the NRF, GENUS, PAST, or the European Union.

## Author Contributions

**Data curation:** Ashley Krüger.

**Formal analysis:** David P. Groenewald, Ashley Krüger.

**Funding acquisition:** David P. Groenewald, Bruce S. Rubidge.

**Investigation:** David P. Groenewald, Ashley Krüger, Michael O. Day, P. John Hancox, Bruce S. Rubidge.

**Resources:** Ashley Krüger, Bruce S. Rubidge.

**Supervision:** Michael O. Day, Cameron R. Penn-Clarke, Bruce S. Rubidge.

**Visualization:** David P. Groenewald, Ashley Krüger.

**Writing – original draft:** David P. Groenewald.

**Writing – review & editing:** David P. Groenewald, Ashley Krüger, Michael O. Day, Cameron R. Penn-Clarke, P. John Hancox, Bruce S. Rubidge.

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
