## [Decision Letter · Decision Letter 0]

7 Dec 2022

PONE-D-22-30091Unique trackway on Permian Karoo shoreline provides evidence of temnospondyl locomotory behaviour.PLOS ONE

Dear Dr. Groenewald,

Thank you for submitting your manuscript to PLOS ONE. After careful consideration, we feel that it has merit but does not fully meet PLOS ONE’s publication criteria as it currently stands. Therefore, we invite you to submit a revised version of the manuscript that addresses the points raised during the review process.

We look forward to receiving your revised manuscript.

Kind regards,

Jörg Fröbisch, Ph.D.

Academic Editor

PLOS ONE

Journal Requirements:

3. We note that you have referenced (ie. Bewick et al. [5]) which has currently not yet been accepted for publication. Please remove this from your References and amend this to state in the body of your manuscript: (ie “Bewick et al. [Unpublished]”) as detailed online in our guide for authors

Reviewers' comments:

Reviewer's Responses to Questions

**Comments to the Author**

1. Is the manuscript technically sound, and do the data support the conclusions?

Reviewer #1: Yes

Reviewer #2: Yes

2. Has the statistical analysis been performed appropriately and rigorously? 

Reviewer #1: N/A

Reviewer #2: N/A

3. Have the authors made all data underlying the findings in their manuscript fully available?

Reviewer #1: No

Reviewer #2: Yes

4. Is the manuscript presented in an intelligible fashion and written in standard English?

Reviewer #1: Yes

Reviewer #2: Yes

5. Review Comments to the Author

Reviewer #1: This is an interesting and well-executed study, describing and interpreting some remarkable body traces attributed to amphibians. Nevertheless, some parts of the manuscript can be improved. Some additional information on coordinates and stratigraphy should be provided. Some close-ups of the circular tracks should be added to the figures. The interpretation of the trace 2 should be expanded, and labels for the single tracks added. The morphology of the body traces shluld be compared with those already described from the Palaeozoic, a new ichnotaxonomy can be proposed. In general, the interpretation in several part can be more tentative. The 3D models could be deposited in an online repository. All the comments are in the attached pdf

Best regards,

Lorenzo Marchetti

Reviewer #2: Dear colleagues,

I have carefully read and reviewed the manuscript, which deals with conspicuous impressions produced by a tetrapod (an amphibian, as the authors thoroughly discuss) on a large palaeosurface from the upper Permian of the Waterford Fm. (Karoo Basin, South Africa). The reported traces are impressive and very interesting, adding valuave knowledge on the locomotion of amphibians (and particularly rhinesuchids) and their palaeoecology. In addition, the authors carried out a very detailed sedimentological study allowing for a detailed palaeoenvironmental reconstruction.

The manuscript is very well writen and the figures are of good quality and very informative. I think this work deserves to be published after a minor revision is undertaken. I have only found minor issues to correct and/or clarify in the text and a few in the figures (for the latter only to show some figured elements better). I annotated all my comments and suggestions in the attached PDF by using the Adobe Reader Comment tools.

The only major point to consider in the text is on the interpretation of the order of impression of some of the traces. It is claimed that the traces described as "Circular impressions" were made before the temnospondyl traces (see comment on page 16 of the manuscript, lines 394-394). However, I can't see this clearly with the given data. The authors show provide further discussion/evidence for this. In any case, I still see this as a minor point because it doesn't change the main message of this research.

There is another point that I didn't comment in the PDF, but that it might be interesting to mention in the manuscript, and in any case only if the authors find this appropriate. As the authors clearly demonstrated, the palaeosurface described was formed in a tidal flat or a lagoon, i.e., a coastal-marine setting. Just as a suggestion, the authors could add a short discussion on the fact that temnospondyl traces are found in such a setting, because usually (though not exclusively) temnospondyls are reported from freshwater settings. Of course, this should not change any content of the manuscript, which I find very good, it would only be adding some palaeoecological remarks on rhinesuchids roaming a coastal-marine setting.

I hope that my review is useful to the authors, for any doubt, I would be glad to give further feedback to the authors. I am looking forward to seeing this work published.

Best regards,

Eudald Mujal (Staatliches Museum für Naturkunde Stuttgart)

6. PLOS authors have the option to publish the peer review history of their article (what does this mean?). If published, this will include your full peer review and any attached files.

Reviewer #1: **Yes: **Lorenzo Marchetti

Reviewer #2: **Yes: **Eudald Mujal

---

## [Author Response · Author response to Decision Letter 0]

3 Feb 2023

Response to Reviewers

Comments from the Journal and the two reviewers are presented below, with our responses in italics. The changes are also shown in a Word document with track changes. Line and page numbers correspond to the annotated version of the manuscript [track changes -> All markup (Show All Revisions Inline)]. All figures have been deleted from the Word documents of the manuscripts.

Response to Journal Requirements:

I have formatted the style of the document to fit with the style requirements of the journal.

I have added the following in lines 177-179 (Pg. 6): “The described study complied with all relevant regulations and the necessary permit (REF: SAH19/13092) was obtained from KwaZulu-Natal Amafa and Research Institute.”

3. We note that you have referenced (i.e. Bewick et al. [5]) which has currently not yet been accepted for publication. Please remove this from your References and amend this to state in the body of your manuscript: (ie “Bewick et al. [Unpublished]”) as detailed online in our guide for authors

We have not referenced Bewick et al. and I am not sure what this is referring to.

The image referred to in Figure 1 is an aerial photograph of the palaeosurface that we took as part of this study using a drone. As such, there is no issue for us to use it. We modified the figure caption to read:

Fig. 1. Geological setting of the Dave Green palaeosurface. A) Simplified geological map of the main Karoo Basin. Position of the study area is indicated. B) Aerial photo of the palaeosurface (taken by AK). C) Stratigraphic log measured along the Rensburgspruit.

Figure 9 (now Fig. 10) included in image of a saltwater crocodile on a tidal flat and we were unable to get the necessary permission to use it. As such, I have modified the figure to include a photo of an impression of an alligator for which I have obtained permission from the photographer, Gale Bishop. I have uploaded the completed Content Permission Form.

Caption of Fig. 10 reads: “Fig. 10. Comparison of Impression 2 from the palaeosurface (A) with a present-day body impression and trail of Alligator mississippiensis on the foreshore at St. Catherines Island, Georgia, United States (B). Photograph courtesy of St. Catherines Island Sea Turtle Program, Gale A. Bishop and modified with permission from [92]. C) A rhinesuchid temnospondyl such as Laccosaurus or Uranocentrodon is probably the tracemaker. Scale bar = 30 cm (A and B).

We have not cited any retracted papers.

Some references that have been added include: 

[61] Marchetti L, Belvedere M, Voigt S, Klein H, Castanera D, Díaz-Martínez I, et al. Defining the morphological quality of fossil footprints. Problems and principles of preservation in tetrapod ichnology with examples from the Palaeozoic to the present. Earth-Sci Rev. 2019;193: 109–145. doi:10.1016/j.earscirev.2019.04.008.

[75] Carmona N, Bournod C, Ponce JJ, Cuadrado D. The Role of Microbial Mats in the Preservation of Bird Footprints: A Case Study from the Mesotidal Bahia Blanca Estuary (Argentina). In: Noffke N, Chafetz H, editors. Microbial Mats in Siliciclastic Depositional Systems Through Time. SEPM Society for Sedimentary Geology; 2011. pp. 37–45. doi:10.2110/sepmsp.101.037

[76] Marty D, Strasser A, Meyer CA. Formation and Taphonomy of Human Footprints in Microbial Mats of Present-Day Tidal-flat Environments: Implications for the Study of Fossil Footprints. Ichnos. 2009;16: 127–142. doi:10.1080/10420940802471027

Reviewers' comments:

Reviewer #1: This is an interesting and well-executed study, describing and interpreting some remarkable body traces attributed to amphibians. Nevertheless, some parts of the manuscript can be improved. Some additional information on coordinates and stratigraphy should be provided. Some close-ups of the circular tracks should be added to the figures. The interpretation of the trace 2 should be expanded, and labels for the single tracks added. The morphology of the body traces shluld be compared with those already described from the Palaeozoic, a new ichnotaxonomy can be proposed. In general, the interpretation in several part can be more tentative. The 3D models could be deposited in an online repository. All the comments are in the attached pdf

Best regards,

Lorenzo Marchetti

Comments from the PDF:

Line 63 (Pg.1): all citations should go in chronological order in the text, unless this journal says otherwise

The reference style has been changed to follow the guidelines of PLOS ONE.

Line 64 (Pg. 2): better "reptiles including dinosaurs" or "dinosaurs and other reptiles"

 Changed “dinosaurs and reptiles” to "dinosaurs and other reptiles"

Line 66 (Pg. 2): I agree on the body traces, but not on the pes and manus footprints because they do not show anatomical detail. Please rephrase. For an extensive discussion about trace preservation, see: Marchetti, L., Belvedere, M., Voigt, S., Klein, H., Castanera, D., Díaz-Martínez, I., ... & Farlow, J. O. (2019). Defining the morphological quality of fossil footprints. Problems and principles of preservation in tetrapod ichnology with examples from the Palaeozoic to the present. Earth-Science Reviews, 193, 109-145.

Changed “remarkably well-preserved” to “remarkable”

Line 69 (Pg.2): [regarding the “preserving”] better "generated by" or similar

 Changed “preserving” to “recording”

Line 85 (Pg. 2): [locality of the site] Please add coordinates

 Added coordinates in line 85 “(S28.967122°, E29.987366°)”

Line 102 (Pg. 3): [Regarding the Waterford Formation] could you provide the thickness of this formation? Also information on its age would be important. And please specify if the contact with the overlying Balfour Formation is conformable

Inserted “Within the study area, the Waterford Formation is at least 140 m thick and is considered to be Wuchiapingian in age (Groenewald 2021).” in lines 121-122

Inserted “conformably overlying” before “lower Balfour” in Line 106

Inserted “(Lopingian)” between “late Permian” and “Daptocephalus” in line 166

Line 142 (Pg. 4): [Heading “Palaeontology”] of the locality

 Changed to “Local palaeontology”

Line 152 (Pg. 4): [attributed to small-to-medium sized dicynodont trackmakers] on which bases?

Added “, based on track morphology and because dicynodonts are the most commonly represented group in the body fossil record,” after “trackmakers” in lines 152-154.

 Deleted “them” in line 154

Line 155 (Pg. 5): [Regarding fossils] here it would be important to know in which formation these fossils were found, and if they were stratigraphically above or below the palaeosurface, if possible

Added the following after “Fig. 3).” in line 158: “Plant fossils below the palaeosurface are generally more fragmentary than those above it. Vertebrate fossils below the palaeosurface are restricted to isolated and fragmented fish bones and scales, whereas vertebrate fossils recovered from the Balfour Formation by us and previous workers on the farm van der Merwe’s Kraal 972 include a partial rhinesuchid amphibian skull (BP/1/7858; c.f. Laccosaurus) and fragmentary dicynodont material.”

Line 194 (Pg. 7): it would be good to make the 3D models available in a public digital repository

 The 3D models have been uploaded to Morphosource

Line 203 (Pg. 8): [Regarding “The Dave Green ichnofossils are”] perhaps better to add "palaeosurface"

 Inserted “palaeosurface” after “Green”

Line 205 (Pg. 8): [Regarding “upper part”] better uppermost or topmost

 Changed “upper” to “uppermost”

Line 217 (Pg. 8): [Regarding “upper”] see comment above. The fact that the palaeosurface is at the boundary between the two formations should be better remarked

Because we have stated uppermost in line 205, I do not think it is necessary to repeat “uppermost” here as well. In the Abstract, we state that the palaeosurface is situated “immediately below the paleaoshoreline of the Ecca Sea” (Line 50), and again under Geological background (line 92), we state that the palaeosurface is situated “immediately below the Ecca-Beaufort contact”

Line 224 (Pg. 8): [Referring to “Palaeocurrent readings”] which is the ripple strike orientation?

Palaeocurrent for asymmetrical ripples is perpendicular to strike, in the direction of the lee side. Have included “, with ripple crest strike orientation of 168-348º,” after “asymmetrical ripples”

Line 229 (Pg. 9): [Referring to “with a strike orientation of 278°”] which is the current direction?

Symmetrical ripples do not indicate palaeocurrent direction. I have changed the strike orientation to “98-278º”

Line 228 (Pg. 9): [Referring to “Several medium-sized depressions”] are these trace fossils or sedimentary structures?

Here we do not specify as it is uncertain. The origin of the depressions is discussed in lines 377-382 and we state (line 380) that they “could have formed as erosional depressions, or possibly footprints,”

Line 233 (Pg. 9): [Referring to “smaller, [sub]circular depressions,”] please specify that these are ichnofossils

Line 233: Changed “circular” to “subcircular”

Changed line 234-236 to read “We consider these to represent trackways with poor morphological preservation and describe them further below under Subcircular depressions.”

Line 240 (Pg. 9): [Referring to “in three groups”] please specify that is based on morphology and arrangement of the traces

 Added “based on the morphology and arrangement of the traces” after “groups”

Line 271 (Pg. 11): [Referring to “15 round depressions”] please provide the diameter

Changed lines 272-275 to read “These depressions, eleven of which are alongside impression 2A and four are alongside impression 2B, have diameters between 33 and 63 mm and some have smooth expulsion rims.”

Line 275 (Pg. 11): [Referring to “lack of morphology”] please explain this in terms of preservation

Lines 275-277 now read: “The footprints have poor morphological preservation (M-preservation grade 0 using the scale of Marchetti et al. (2019a)) that makes it difficult to distinguish manus from pes.”

Line 279 (Pg. 11): [Referring to “A similar distance separates the posterior from the anterior pair of footprints in impression 2A.”] how can you tell? Impression 2A has 11 associated footprints. Perhaps specify that a similar distance can be measured between supposed posterior and anterior pairs taking into account progression while generating the body trace, which in fact results longer than 2B. You should add numbers to the pairs in the figure and refer to them (also provide measurements of gleno-acetabular distance, stride, pace angulation, trackway width)

Included “(RP3-RM5 and LP3-LM4)” after “respectively” in line 279

Lines 279-282 now read: “A similar distance separates the supposed posterior and anterior pairs of footprints, taking into account progression while generating the body trace (LP1/RP1-LM2/RM2 or LP2/RP2-LM3/RM4), in impression 2A (Fig. 7).” 

Fig. 7 has been modified following the suggestions, including adding numbers to the impressions and footprints and marking the gleno-acetabular distance.

Line 300 (Pg. 12): [Referring to “swim”] interpretation later

 Deleted “swim”

Line 302 (Pg. 12): [“Circular depressions”] please add close-ups of these structures, they are not visible in the current figures

Changed “circular depressions” to “Subcircular depressions”

Replaced “closely grouped” with “Subcircular” in Line 241

Modified the paragraph describing these and have also added a new figure, Figure 8.

Lines 303-313 now read: “Several groupings of smaller, subcircular and blob-shaped depressions occur across the palaeosurface. Three such groupings, indicated in Fig. 5 as “Smooth area with depressions” since the area surrounding the depressions is often smooth and not rippled, are: 1) a “corridor” ~0.9 m wide that crosses the surface from the western to eastern side (Fig. 4G and Fig 8A-E); 2) a concentration just north of the present-day island; and 3) a higher density of these depressions preserved in the northwestern part of the surface (Fig. 8). The depressions vary in size and shape with diameters ranging from 10–15 cm. The bottom of many of the depressions is sculpted with asymmetrical ripples and little-to-no morphological details are preserved.”

Lines 332-333 (Pg. 13): [“and lack the typical sinuous shape normally associated with fish trails”] this should go in the discussion

I have deleted this half of the sentence. In the discussion it already says in lines 477-478 “While most of the traces are straight and lack the sinusoidal pattern characteristic of Undichna,”

Added “of” to the sentence between “most” and “the” in lines 477-478

Line 364 (Pg. 14): [Referring to the second “which”] with?

 This is an error and I have deleted “which”

Line 365 (Pg. 14): [“indicate that the palaeosurface was submerged at the time that the traces were made”] please be more tentative here, the Undichna assignment is not certain and the surface may have been flooded again

The assignment to Undichna is also quite certain for some of the linear traces, e.g. LT1 and the Undichna unisulca trace shown in Fig. 9I.

Changed to: “indicate” to “suggests”

Line 366 (Pg. 14): Replace “traces” with “large impressions”

 Done

Line 383 (Pg. 15): [Referring to “The smoother, non-rippled areas of the surface are possibly where a biofilm or microbial mat was growing on the sand in a shallow water environment…”] is it possible that these areas were simply not submerged or deep enough for the ripple formation? Interestingly, the supposed tracks are in these areas. Also, if microbial mats were present, there should be sedimentological evidence such as elephant skin structures

Areas with smooth, non-rippled surfaces adjacent to rippled ‘erosional patches’ are commonly observed on tidal flats (Cuadrado et al. 2011) and we feel we have adequately shown this with the photographs in Figure 4.

CUADRADO, D. G., CARMONA, N. B. & BOURNOD, C. 2011. Biostabilization of sediments by microbial mats in a temperate siliciclastic tidal flat, Bahia Blanca estuary (Argentina). Sedimentary Geology 237(1–2), 95–101. doi: 10.1016/j.sedgeo.2011.02.008

We have also added the following (Lines 387-390): “In parts of the palaeosurface, the surface weathers with distinctive quadrangular chips. These resemble ‘mat chips’ produced by the erosion of modern microbial mats by waves and currents (Cuadrado et al. 2011) and further support the presence of a microbial mat on the surface.”

Line 390-391 (Pg. 15): about trace preservation in microbial mats, see: Marty, D., Strasser, A., & Meyer, C. A. (2009). Formation and taphonomy of human footprints in microbial mats of present-day tidal-flat environments: implications for the study of fossil footprints. Ichnos, 16(1-2), 127-142.

I have now cited this paper in lines 392 and 443-446

Line 398 (Pg. 16): you can consider doing a new ichnotaxonomy of the large impressions. Also, a morphological comparison with the other supposed Paleozoic amphibian body impressions should be done

Because there is, at this stage, not a physical cast of the specimen that can be deposited in a collection and easily accessed for study by others, we have decided not to do the full ichnotaxonomy yet. We have tried to make casts but because the structure is large and very shallow it was not possible to make a cast which reliably reflected the morphology. A full morphological comparison is also something that will be done at a later stage.

Lines 429-430 (Pg. 17): add the ichnospecies name to both these ichnogenera [Hermundurichnus and Sauropleura]

Done

Lines 429: please indicate in which instances your traces differ from these ichnospecies

The most notable difference, size, has already been stated in lines 435-436: “All these examples are, however, as much as 10 times smaller than the impressions reported here.”

Line 464 (Pg. 18): [dicynodont therapsids as most likely trackmakers] please add an explanation for this, and references for dicyndont tracks found in the Balfour Formation

Modified the paragraph so it now reads: “The morphology of the traces provides little information on the identity of the trackmaker but the most likely candidates are dicynodont therapsids. Trackways made by groups of dicynodonts have been reported previously from the Karoo (de Klerk 2002; MacRae 1990), and they are the most common animals of the required body size recorded by body fossils from the area.”

Line 518 (Pg 20): [“The fact that many of the impressions are isolated and have no connecting trackways or swim traces (e.g., the gap between Impressions 2 and 3) is interpreted as evidence for subaqueous activity (Hasiotis et al. 2007)”] it can also be a preservational bias due to different mechanical properties of the substrate in different areas at the time of the impression

Added the following two sentences (lines 522-525): “An alternative interpretation is that the gaps resulted from a preservational bias due to different mechanical properties of the substrate in different areas at the time of registration. However, because of the lack of disturbance or deformation of surface between impressions, especially to the ripple crests, this is considered less likely.”

Line 535 (Pg 21): [“Changes in the footprint trackway associated with impressions 2A and 2B could therefore reflect a shift from bottom walking, with a similar but more variable stride length to terrestrial pedestrianism in 2A, after which the trackmaker pushed up and floated a short distance before settling on the bottom – hence only leaving four footprints associated with Impression 2B.”] or it was all walking and the other tracks were just not preserved

Line 539: Inserted “It is also possible that both impressions were created through walking but that some of the tracks were just not preserved.” after “Impression 2B.”

Line 563 (Pg.22): [“However, the absence of well-defined ripples in most of the body impressions in conjunction with the swim traces indicates that the current was no longer strong enough to overprint the traces with new ripple marks before the surface was buried and preserved by the overlying strata.”] or the water was too shallow/absent

Added “The lack of ripple marks could also suggest that the water was too shallow or absent at the time registration, although, if this were the case, we would expect to see more footprints associated with the traces.” at the end of the paragraph (Line 569-571)

585-596: this part is very interpretive, consider editing, reducing or removing

I have edited this paragraph,and it now reads (Line 572-585): “Some of the traces preserve sharp turns or changes in direction, e.g., Impressions 3, 4, and 5. No observable footprint impressions or disturbances to the adjacent rippled surface are associated with the turns as observed in other amphibian swimming traces (Turek 1989), and the slightly enlarged area anterior to the tail implies that the hind limbs were tucked against the body. It is possible that the animal used its front legs to help steer in a manner similar to younger crocodiles (Seebacher et al. 2003) and that they did not touch the substrate. Another possibility is that the animal did not use its legs at all while swimming, as seen in larger crocodiles (Seebacher et al. 2003), but they were used only when the animal touched the substrate, or while bottom walking (Farlow et al. 2018b; Mujal & Schoch 2020). While a direction of travel cannot be determined for LT1, it is worth noting that Impressions 1-4 run almost parallel to LT1 (Fig. 5). If LT1 is taken to have been travelling north-south before turning and heading west-east, this could suggest that the tracemaker was actively following the fish, and that the traces record some hunting behaviour. Furthermore, the point where the smooth trace and LT1 are closest together, is near the sinuous section, i.e., at which point the fish put in a burst of speed to escape.”

I also edited lines 325-335 to follow the same direction of travel, i.e., start in the north. Figure 9 has also been updated to reflect these changes. This paragraph now reads: “The first linear trace (LT1) consists of a paired trace that can be followed southwards from the northwestern edge of the surface, where it either overprints or is overprinted by Impression 1 (Fig. 9A and B). It makes a sharp turn just south of Impression 7 (Fig. 9C and D) and can be followed eastwards across the surface to the eastern margin of the palaeosurface (Fig. 9E). The individual traces are ~1 cm wide and the paired traces are ~35 cm apart. Except for a short, sinuous section near Impression 4, the trails are relatively straight. In the sinuous section (Fig. 9F), the trails have a wavelength of 36–39 cm and an amplitude of 5 cm.”

Figure 4G: add close-ups

Created a new figure, Fig. 8, which shows the subcircular depressions better.

Fig 7: [scale] better in cm

The figure has been modified and now includes a scale bar of 50 cm

Fig 7: add track numbers and body trace letters

Done

Reviewer #2: Dear colleagues,

I have carefully read and reviewed the manuscript, which deals with conspicuous impressions produced by a tetrapod (an amphibian, as the authors thoroughly discuss) on a large palaeosurface from the upper Permian of the Waterford Fm. (Karoo Basin, South Africa). The reported traces are impressive and very interesting, adding valuable knowledge on the locomotion of amphibians (and particularly rhinesuchids) and their palaeoecology. In addition, the authors carried out a very detailed sedimentological study allowing for a detailed palaeoenvironmental reconstruction.

The manuscript is very well written and the figures are of good quality and very informative. I think this work deserves to be published after a minor revision is undertaken. I have only found minor issues to correct and/or clarify in the text and a few in the figures (for the latter only to show some figured elements better). I annotated all my comments and suggestions in the attached PDF by using the Adobe Reader Comment tools.

The only major point to consider in the text is on the interpretation of the order of impression of some of the traces. It is claimed that the traces described as "Circular impressions" were made before the temnospondyl traces (see comment on page 16 of the manuscript, lines 394-394). However, I can't see this clearly with the given data. The authors show provide further discussion/evidence for this. In any case, I still see this as a minor point because it doesn't change the main message of this research.

There is another point that I didn't comment in the PDF, but that it might be interesting to mention in the manuscript, and in any case only if the authors find this appropriate. As the authors clearly demonstrated, the palaeosurface described was formed in a tidal flat or a lagoon, i.e., a coastal-marine setting. Just as a suggestion, the authors could add a short discussion on the fact that temnospondyl traces are found in such a setting, because usually (though not exclusively) temnospondyls are reported from freshwater settings. Of course, this should not change any content of the manuscript, which I find very good, it would only be adding some palaeoecological remarks on rhinesuchids roaming a coastal-marine setting.

I hope that my review is useful to the authors, for any doubt, I would be glad to give further feedback to the authors. I am looking forward to seeing this work published.

Best regards,

Eudald Mujal (Staatliches Museum für Naturkunde Stuttgart)

Regarding the marine setting – The lower Ecca Group is generally accepted as having been deposited under marine conditions, whereas the upper Ecca Group was deposited under brackish-to-fresh water. As such, we have changed “marine” in line 365 to “water” and added the following sentence after “(de Gibert & Benner 2002).” in line 366: “The water was likely brackish-to-fresh based on the conclusions of several studies using trace elements from the upper Ecca Group (Muntingh 1997; Veevers et al. 1994; Zawada 1988).”

Comments from the PDF:

Line 49 (Pg. 1): [“a late Permian”] “an upper”

We have left this unchanged. The “late” applies as a chronological rather than a stratigraphic term. The palaeosurface is not considered upper Permian stratigraphically, since it is at the base of the Beaufort Group in KwaZulu-Natal Province. Because the palaeosurface is late Permian in age, we have used late Permian.

Line 55 (Pg. 1): Replace “marks” with “grooves”

 Changed to: “The sinuous shape of some of the traces”

Line 56 (Pg. 1): Change trackmaker to tracemaker

 Done

Line 78 (Pg. 2): [“photogrammetry scans”] I think this should rather be "surface scans".

 Changed to "surface scans"

Line 79 (Pg.2): [“trackmaker”] Since most of the impressions you are describing are not tracks (understood as the imprints produced by autopodia), I would write "tracemaker" instead of "trackmaker". In fact, I see that later in the text you use "tracemaker".

 Done. Also changed in line 80

Line 89 (Pg. 3): replace “late” with upper

 Left unchanged. See response for line 49

Line 101 (Page 3): [Fig 2 caption] Replace “manuscript” with “text”

 Done

Line 105 (Pg. 3): “f” lowercase in “Formation”

 Done

Line 120 (Pg. 4): replace “suggest” with “suggested”

 Changed “suggest” to “showed”

Line 138-141 (Pg. 4): [“The depositional environment for the Balfour Formation in the northeastern main Karoo Basin has been interpreted as high load meandering river environments (Botha & Linström 1977, 1978; Green 1997; Groenewald 2021, 1989, 1990; Groenewald et al. 2022; Johnson et al. 2006; Muntingh 1989, 1997).”] This sentence is the nearly the same as the anterior one. Please delete one of the two.

 Deleted the second sentence

Line 137 (Pg. 4): Replace “environments” with “systems”

Done

Line 157 (Pg. 5): [D.G.] insert “P.”

 Changed to DPG

Line 163 (Pg. 5): delete “c.f.”

 Unchanged. The identity of BP/1/7858 is tentative

Changed “an almost complete rhinesuchid” to “partial rhinesuchid” as BP/1/7858 comprises only the posterior portion of the skull and the snout is missing.

Line 165 (Pg. 5): [“KZN”] Is this KwaZulu-Natal? If so, write this instead of the abbreviation (not used anywhere else in the text).

 Changed to “KwaZulu-Natal

Table 1: Write the facies codes in italics.

 Done

Table 1: insert “,” between “deposition” and “e.g.” [in the interpretation of “Sm”]

 Done

Table 1: [“couple” in the interpretation of “Sm”] Should this be "could"?

Corrected to “could”

Table 1: [“Fl” in description of “Fm”] also in italics

 Done

Line 181 (Pg. 7): Change “which” to “that”

 Done

Line 186 (Pg. 7): [“photogrammetry”] I think this should rather be "surface scanning".

 Changed “white-light source photogrammetry” to “surface scanning”

Lines 183-187 “Consequently, we used high-resolution scanning to digitise and accurately record the palaeosurface. Combining surface scanning and aerial images, we were able to accurately portrait the surface trackway.” have been combined and rewritten and now reads (Lines 183-185) “Consequently, we combine high-resolution surface scanning and aerial images to digitise and accurately record the surface.”

Line 194 (Pg. 7): Above you mentioned photogrammetry was performed, but here it is not mentioned, I guess before you meant surface scanning. Please either add information on any photogrammetric models you did or remove the references to photogrammetry.

 All references to photogrammetry have been removed. 

 Added “and ParaView v. 5.10.1 (https://www.paraview.org/)” at the end of line 200

Line 226 (Pg. 9): [“Gyrochorte-like invertebrate traces are present on the northwestern part of the surface (Fig. 4C and D).”] I can't identify the Gyrochorte-like traces in the cited subfigures. Could you please point them (e.g., with arrows) in the photographs?

I have modified the photo in 4D and placed arrows to show the invertebrate traces

Line 255 (Pg. 10): Upper case (as done elsewhere for "Impression/s #number").

 Done. Also in Line 257

Caption Fig. 6: [“In parts 1–5: scale] Should this be 1-6? Otherwise provide the length of the scale bar in 6

 Corrected to “1–6”

Caption Fig.7 First letter capitalized [for “impression”] (as done in the text).

 Done

Table 2: I would put both length and width in mm, so that they both are in the same units.

Put them both into cm in the table. In the text the measurements remain unchanged.

Table 2: If possible/relevant, could you add also dimensions for Impression 6? I suggest this mainly because it is shown in Fig. 6.

We do not have the measurements for this one as the measurements were taken off the models produced by the scans and Impression 1 and 6 were not scanned.

Line 284 (Pg.11): Delete “-“ in “Trace-maker”

 Done. Also done elsewhere in the text

Line 291 (Pg. 12): Do you mean "faint" here?

 Replaced “feint” with “faint”

Line 304 (Pg. 12): Insert “of” before “such”

 We have left unchanged as “three of such” does not make sense.

Line 303 has been changed and now reads: “Several groupings of smaller, subcircular and blob-shaped depressions occur across the palaeosurface.”

Caption Fig. 8[ now 9]: Delete “pairedlarge”

 Done

Caption Fig. 8[now 9]: Add the length of the hammer here please (e.g., Hammer (30 cm long) for scale). And move this to the end of the caption, since the hammer appears in several photos.

 Done

Caption Fig. 8 [now 9]: The arrows are not very visible, they should be (slightly) larger and/or in another colour (e.g., white, or with black outline and white infill).

 Arrows have been modified in Figure 9I and are now more visible

Inserted new picture in Figure 9 (9B) which shows the trace crossing Impression 1.

Line 357 (Pg. 14): Add an “h” to “Undicna”

 Done

Line 364 (Pg. 14): Delete second which in sentence

 Done

Line 390-391 (Pg. 15): [comment] See also Marty et al. (2009: Ichnos), who discussed on the different preservations of tracks in microbial mats.

Marty, D., Strasser, A., Meyer, C.A., 2009. Formation and taphonomy of human footprints in microbial mats of present-day tidal-flat environments: implications for the study of fossil footprints. Ichnos 16, 127–142.

These lines have been modified and now read: “Tracks and traces require the substrate to be ‘just right’ in order to be preserved (Falkingham et al. 2014). Under certain conditions, the presence of microbial mats have been shown to favour or enhance the preservation of footprints (Carmona et al. 2011; Marty et al. 2009) and microbial mats are commonly associated with fossil trackways, including in the Karoo Basin (Marsicano et al. 2014; Sciscio et al. 2016, 2020; Smith 1993).”

Line 393 (Pg 15): Add “Basin” after “Karoo”

 Done. Also added “main” before Karoo in the same line.

Line 398 (Pg. 16): [“impressions”] italics

 The headings are now formatted according to the style of PLOS ONE (14 pt and Bold)

Line 404-405 (Pg. 16): [“Early Jurassic”] Change “Early” for “Lower”

 Done

Figure 9 [now 10]: It might be better to add a letter (B) in the photograph of Impression 2, separating it from the subfigure A. In this sense, B should be then C.

 Done

Caption Figure 9 [now 10]: [“impression”] Upper case

 Done

Figure 9 [now 10]: As a whole, I like this figure a lot! I would add a scale bar in B [C in the updated figure].

 C is not drawn to scale

Caption Figure 9 [now 10]: [“trace-maker”] delete “-”

 Done

Line 421 (Pg. 16): Change “trackmaker” to “tracemaker”

 Done

Line 422 (Pg. 16): Replace “late” with “upper”

We have left this unchanged. The “late” applies as a chronological rather than a stratigraphic term. The palaeosurface is not considered upper Permian stratigraphically, since it is at the base of the Beaufort Group in KwaZulu-Natal Province. Because the palaeosurface is late Permian in age, we have used late Permian.

Line 440 (Pg. 17): [“trace maker”] Without space between the two words.

 Done

Lines 447 (Pg. 17): As before, please check also Marty et al. (2009), as well as Carmona et al. (2011), who also focused on the taphonomy of tracks on microbial mats.

Marty, D., Strasser, A., Meyer, C.A., 2009. Formation and taphonomy of human footprints in microbial mats of present-day tidal-flat environments: implications for the study of fossil footprints. Ichnos 16, 127–142.

Carmona, N., Bournod, C., Ponce, J.J., Cuadrado, D., 2011. The role of microbial mats in the preservation of bird footprints: a case study from the mesotidal Bahia Blanca estuary (Argentina). In: SEPM Special Publications, 101, pp. 37–45. https://doi.org/ 10.2110/sepmsp.101.037.

These references have been included and lines 444-446 have been modified to read: “The lack of morphology in the associated footprints could also be the result of the interplay between the algal mat and substrate properties (Carmona et al. 2011; Marty et al. 2009). Alternatively, post-registration growth of the microbial mat on the substrate could have resulted in the loss of morphological detail over time (Carmona et al. 2011; Cuadrado et al. 2012; Gerdes 2007; Marty et al. 2009; Reineck & Singh 1975, 1980; Schieber 1998; Selley 1985).”

Line 459 (Pg. 19): [“liquefied”] change to “liquified”

 Both spellings are correct. Left unchanged

Line 462-463 (Pg. 18): [“This indicates that these trackways were probably made before the temnospondyl traces.”] I am not totally convinced. With what I see in Fig. 4G, the "corridor" of footprints is not smoothed/overprinted by any structure, and it seems that Impressions 2 and 3 are connected by a smooth trace, as shown in Fig. 5. Therefore, I would expect that the depressions in the corridor at the height of Impressions 2 and 3 should be more smoothed than the others (i.e., overprinted by the temnospondyl tracemaker). In this sense, the trackways you interpret should have been imprinted after the temnospondyl traces. Maybe it is a matter that I don't see this well in the figures. If you have a close up of this region, it would be worth it to figure it.

The area is now better figured in a new figure (Figure 8)

It is also not clear that the trackway has overprinted a swim trace, as one cannot be picked up on either side of the “corridor”. Our main argument for the order of the traces i.e., that the trackway is older, is because there are ripples in some of the footprints but not in the amphibian body traces. This is explained in Lines 459-462. It is not definite and for that reason we have said that the trackways were probably made before the impressions (line 462).

Line 477 (Pg. 19): Insert “of” after “most”

 Done

Line 482 (Pg. 19): Do some of the more linear traces align with the palaeocurrents? If so, this could support the hypothesis of a stick drifting and dragging along the sediment. It would be worth it to mention the relationship in orientation between these traces and the palaeocurrents.

Modified the sentence to read: “It is also possible, however, that some of the more linear examples such as LT2 and LT3, which have a similar orientation to the palaeocurrent direction, were created by a stick drifting and dragging along the sediment, thereby forming a pseudo trace fossil (R. Gess, pers. comm. 2015).”

Line 518 (Pg. 20): Please see comment in page 16 (lines 393-394) above. Could it be that the traces aren't connected because the circular structures interpreted as trackways overprinted the connection area?

That may be the case between Impressions 2 and 3, but what about between Impressions 1 and 2, and 5 and 6 where there are areas with no subcircular depressions.

Line 530 (Pg. 21): Please consider to cite here Farlow et al. (2018b) and Mujal & Schoch (2020), who also discussed this fact.

 Done

Line 553 (Pg.22): Change “trackmaker” to “tracemaker”

 Done

Lines 558, 559, and 572 (Pg. 22): Capitalise “I” in “impression”

 Done

Additional changes:

Line 67: Changed “Estcourt district” to “uThukela District”

Line 93: changed “recording” to “, which records”

Line 107: Inserted “Formation” after “Balfour” and moved the “)” from behind “Estcourt” to after the citations

Line 151: changed “P” to “p” in “palaeosurface”

302 (Pg. 12): Here, and throughout the text, changed “circular depressions” to “subcircular depressions” or “subcircular to blob-shaped”

Lines 378-379: Deleted “similar to the oscillatory ripples from other parts of the section (Groenewald 2015, 2021), but”

Line 396: Corrected spelling of “mineralization” to “mineralization”

Lines 487-491: Inserted “Because of this, and in contrast to many present-day and fossil examples of crocodilian and temnospondyl trackways, which typically comprise a trackway with a central tail or belly drag (Cisneros et al. 2020; Farlow et al. 2018a; Marsicano et al. 2014; Milàn & Hedegaard 2010), the impressions and traces on the Dave Green palaeosurface lack associated footprints or scratch marks, with the exception of Impression 2.”

Lines 541-547: Moved paragraph: “Impressions 2 and 3 are relatively close together and appear to follow on from one another (Figs. 2 and 5). As such, we consider these to have been formed by the same individual. The direction of movement can be determined by looking at the tail to body direction. For example, impression 2 indicates an animal that was moving in a south-easterly direction, as evinced by the tail thickening in the body direction as well as the more pronounced expulsion rim along the posterior margin of some of the footprints, indicating the direction that the animal pushed against the substrate.” earlier in the text. It is now between lines 511-517. 

Lines 515-516: Changed “slightly bulged area behind” to “more pronounced expulsion rim along the posterior margin of”

Line 623: Changed “for” to “during” and “goes” to “go”

Line 624: inserted “We are grateful to Academic Editor Prof. Jörg Fröbisch for handling our manuscript, and to Dr Lorenzo Marchetti and Dr Eudald Mujal for their constructive reviews.”

Line 626-628: Changed to read:” Opinions expressed and conclusions arrived at, are those of the authors only and do not necessarily reflect those of are not necessarily to be attributed to the NRF, the CoE-PalGENUS, or to PAST, or the European Union.”

Line 636: Changed “JH” to “PJH”

---

## [Editor Report · Decision Letter 1]

14 Feb 2023

Unique trackway on Permian Karoo shoreline provides evidence of temnospondyl locomotory behaviour.

PONE-D-22-30091R1

Dear Dr. Groenewald,

We’re pleased to inform you that your manuscript has been judged scientifically suitable for publication and will be formally accepted for publication once it meets all outstanding technical requirements.

Kind regards,

Jörg Fröbisch, Ph.D.

Academic Editor

PLOS ONE

Additional Editor Comments (optional):

I carefully read through the revised version of your manuscript and consider all points raised by the reviewers as addressed to my satisfaction.
---

## [Editor Report · Acceptance letter]

2 Mar 2023

PONE-D-22-30091R1 

Unique trackway on Permian Karoo shoreline provides evidence of temnospondyl locomotory behaviour. 

Dear Dr. Groenewald:

I'm pleased to inform you that your manuscript has been deemed suitable for publication in PLOS ONE. Congratulations! Your manuscript is now with our production department. 

Kind regards, 

on behalf of

Prof. Jörg Fröbisch 

Academic Editor

PLOS ONE